# Grounding inductive biases in natural images: invariance stems from variations in data

**Diane Bouchacourt**,* **Mark Ibrahim**,* **Ari S. Morcos**
Facebook AI Research
{dianeb,marksibrahim,arimorcos}@fb.com

## Abstract

To perform well on unseen and potentially out-of-distribution samples, it is desirable for machine learning models to have a predictable response with respect to transformations affecting the factors of variation of the input. Here, we study the relative importance of several types of inductive biases towards such predictable behavior: the choice of data, their augmentations, and model architectures. Invariance is commonly achieved through hand-engineered data augmentation, but do standard data augmentations address transformations that explain variations in real data? While prior work has focused on synthetic data, we attempt here to characterize the factors of variation in a real dataset, ImageNet, and study the invariance of both standard residual networks and the recently proposed vision transformer with respect to changes in these factors. We show standard augmentation relies on a precise combination of translation and scale, with translation recapturing most of the performance improvement—despite the (approximate) translation invariance built in to convolutional architectures, such as residual networks. In fact, we found that scale and translation invariance was similar across residual networks and vision transformer models despite their markedly different *architectural* inductive biases. We show the training data itself is the main source of invariance, and that data augmentation only further increases the learned invariances. Notably, the invariances learned during training align with the ImageNet factors of variation we found. Finally, we find that the main factors of variation in ImageNet mostly relate to appearance and are specific to each class.

## 1 Introduction

A dataset can be described in terms of natural factors of variation of the data: for example, images of objects can present those objects with different poses, illuminations, colors, etc. Prediction consistency of models with respect to changes in these factors is a desirable property for out-of-domain generalization [5, 25]. However, state-of-the-art Convolutional Neural Networks (CNNs) struggle when presented with "unusual" examples, e.g. a bus upside down [2]. Indeed, CNNs lack robustness not only to changes in pose, but even to simple geometric transformations such as small translations and rotations [4, 15].

Invariance to factors of variation can be learned directly from data, built-in via architectural inductive biases, or encouraged via data augmentation (Fig. 1A). Our goal with this work is to explore the relative impact of these three factors on the trained model's learned invariances and performance. As of today, data augmentation is the predominant method for encouraging invariance to a set of transformations. Yet, even with data augmentation models fail to generalize to held out objects and to learn invariance. For example, [15] found that augmenting with rotation and translation does not lead to invariance to the very same transformations during testing. A complementary research direction ensures a model responds predictably to transformations, using group equivariance theory (see Cohen

---

*equal contribution, a coin was flipped

35th Conference on Neural Information Processing Systems (NeurIPS 2021).

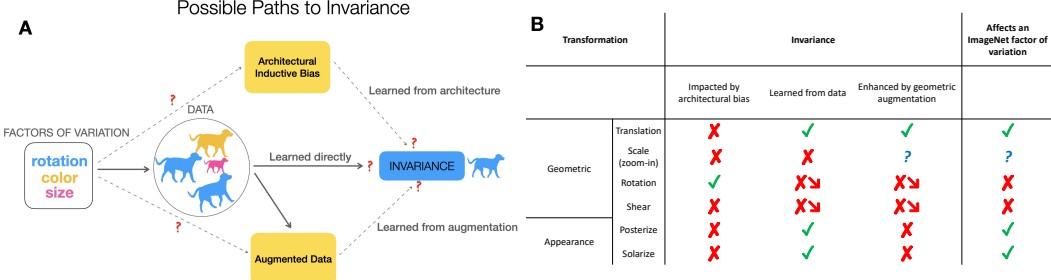

Figure 1: **Grounding invariance in factors of variation**. In Fig. A, we illustrate possible paths to learning invariance: architectural inductive bias, directly from data, and through data augmentation. In Fig. B, we summarize our findings about invariance. A red arrow means invariance decreases, a question mark means results do not allow for a clear answer.

and Welling [9], Cohen et al. [10] among others). Provably invariant models have limited large-scale applications as they require a priori knowledge of the underlying factors of variation [9, 17]. Recent work tackles automatic discovery of symmetries in data [6, 12, 20, 39]. However, these methods have mostly been applied to synthetic or artificially augmented datasets which are not directly transferable to real data settings, and can even hurt performance when transferred [12]. Thus, we aim to identify the factors of variation of a real image dataset and to understand whether such equi/invariant models would be a relevant choice.

One way to characterize the consistency of a model's response is by measuring its equivariance or invariance to changes of the input. A model, $f$, is equivariant to a transformation $T_\theta$ of an input $x$ if the model's output transforms in a corresponding manner via an output transformation $T'_\theta$, i.e. $T'_\theta(f(x)) = f(T_\theta(x))$ for any $x$. Invariance is a special type of equivariance, where the model's output is the same for all transformations, i.e. $T'_\theta = I$. Throughout this work, we will use for $f$ the penultimate layer of a Resnet18 [22] or a vision transformer [14] trained on ImageNet. To understand whether CNNs and other state-of-the-art models are invariant to changes in the data factors of variation, one needs explicit annotations about such factors. While this is trivial for synthetic datasets with known factors, identifying the factors of variations in real datasets is a complex task. As such, prior work turned to synthetic settings to show that knowing the underlying factors improves generalization [8, 26].

Here, we take a first step towards understanding the links between data augmentations and factors of variation in natural images, in the context of image classification. We do so by carefully studying the role of data augmentation, architectural inductive biases, and the data itself in encouraging invariance to these factors. We primarily focus on ResNet18 trained ImageNet as a benchmark for large-scale vision models [13, 22], which we also compare to the recently proposed vision transformer (ViT) [14]. While previous works study the invariance properties of neural networks to a set of transformations [15, 23, 25, 27, 38], we ground invariances in ImageNet factors of variations. We make the following contributions (summarized in Fig. 1):

- *What transformations do standard data augmentation correspond to?* In Sec. 2, we demonstrate that the success of the popular random resized crop (RRC) augmentation amounts to a precise combination of translation and scaling. To tease out the relative role of these factors, we decomposed RRC into separate augmentations. While neither augmentation alone was sufficient to fully replace RRC, we observed that despite the approximate translation invariance built into CNNs, translation alone is sufficient to improve performance close to RRC, whereas the contribution of scale was comparatively minor.

- *What types of invariance are present in ImageNet trained models? Do these invariances derive from the data augmentation, the architectural bias or the data itself (Figure 1A)?* In Sec. 3, we demonstrate that when invariance is present, it is primarily learned from data independent of the augmentation strategy used with the notable exception of translation invariance which is enhanced by standard data augmentation. We also found that architectural bias has a minimal impact on invariance to the majority of transformations.

- *Which transformations account for intra-class variations in ImageNet? How do they relate to the models' invariance properties discovered in Sec. 3?* In Sec. 4, we show that appearance transformations, often absent from standard augmentations, account for

| Method | Top-1 $\pm$ SEM | | $\beta$ | Top-1 $\pm$ SEM |
|---|---|---|---|---|
| `RandomResizedCrop` (RRC) | **70.05 $\pm$ 0.1** | | 0.1 | 69.32 $\pm$ 0.0 |
| `RandomSizeCenterCrop` | 67.84 $\pm$ 0.1 | | 0.5 | **70.17 $\pm$ 0.1** |
| `FixedSizeRandomCrop` | 67.93 $\pm$ 0.0 | | 1 ($\sim$ RRC) | **70.11 $\pm$ 0.0** |
| `T.(30%)` | 69.14 $\pm$ 0.0 | | 2 | 69.52 $\pm$ 0.1 |
| `T.(30%) + RandomSizeCenterCrop` | 69.30 $\pm$ 0.0 | | 3 | 68.79 $\pm$ 0.0 |
| `T.(30%) + RandomSizeCenterCrop` w/o a.r. | 69.20 $\pm$ 0.1 | | 10 | 63.74 $\pm$ 0.1 |
| `FixedSizeCenterCrop` (no augmentation) | 63.49 $\pm$ 0.1 | | | |

(a) Using different training augmentations used, w/o a.r. stands for without aspect ratio change, T. stands for Translation.

(b) Varying the shape of the distribution over $s$. RRC stands for `RandomResizedCrop`.

Table 1: ImageNet validation Top-1 accuracy $\pm$ SEM (standard error of the mean).

intra-class changes for factors of variation in ImageNet. We found training enhances a model's natural invariance to transformations that account for ImageNet variations (including appearance transformations), and decreases invariance for transformations that do not seem to affect factors of variation. We also found factors of variation are unique per class, despite common data augmentations applying the same transformations across all classes.

Our results demonstrate that the relationship among architectural inductive biases, the data itself, the augmentations used, and invariance is often more complicated than it may first appear, even when the relationship appears intuitive (such as for convolution and translation invariance). Furthermore, invariance generally stems from the data itself, and aligns with the data factors of variations. By understanding both which invariances are desirable and how to best encourage these invariances, we hope to guide future research into building more robust and generalizable models for large scale, realistic applications. Code to reproduce our results is in supplementary material.

## 2 Decomposing the Random Resized Crop Augmentation

Data augmentation improves performance and generalization by transforming inputs to increase the amount of training data and its variations [28]. Transformations typically considered include taking a crop of random size and location (random resized crop), horizontal flipping, and color jittering [22, 32, 33]. Here, we focus on random resized crop (denoted `RandomResizedCrop`; RRC) that is commonly used for training ResNets [2].

For an image of width $W$ and height $H$, `RandomResizedCrop` (1) samples a scale factor $s$ from a uniform distribution, $s \sim U(s_-, s_+)$ and an aspect ratio $r \sim U(\ln r_-, \ln r_+)$ (2) takes a square crop of size $\sqrt{sHWr} \times \sqrt{sHW/r}$ in any part of the image (3) resizes the crop, typically to $224 \times 224$ for ImageNet. Thus, the area of an object selected by the crop is randomly scaled proportional to $1/s$, which encourages the model to be scale invariant. The crop is also taken in any location of the image within its boundaries, which is equivalent to applying a translation whose parameters depend on the percentage of the area chosen for the crop. However, the way translation and scale interact remains obscure. In this section, we contrast the role of translation and scale in RRC and analyze the impact of the parameters used to determine these augmentations.

### 2.1 The gain of `RandomResizedCrop` is largely driven by translation rather than scale

To study the role of both the scaling and translation steps, we separate `RandomResizedCrop` into two component data augmentations:

- `RandomSizeCenterCrop` takes a crop of random size, always at the center of the image. The distribution for scale and aspect ratio are the same as those used in `RandomResizedCrop`. This augmentation impacts scale, but not translation.
- `FixedSizeRandomCrop` takes a crop of fixed size ($224 \times 224$) at any location of the image (the image is first resized to 256 on the shorter dimension). This augmentation impacts translation, but not scale.

---

[2]We follow the procedure of `RandomResizedCrop` as implemented in the PyTorch library [29] https://pytorch.org/vision/stable/_modules/torchvision/transforms/transforms.html.

As with RRC, both of these transformations can remove information from the image (for example, translation can shift a portion of the image out of the frame whereas zooming in will remove the edges of the image), but neither augmentation can fully reproduce the effect of RRC by itself.

We train ResNet18 on ImageNet and report results on the validation set as in commonly done (e.g. in Touvron et al. [33]) since the labelled test set is not publicly available. `FixedSizeCenterCrop` corresponds to what is usually done for augmenting validation/test images, i.e. resize the image to 256 on the shorter dimension and take a center crop of size 224. We apply `FixedSizeCenterCrop` during all evaluation steps and refer to it as "no augmentation". Training details are in Appendix A.

**RCC combines scale and translation in a precise manner.** Table 1a shows that `RandomResizedCrop` performs best, with $70.05\%$ Top-1 accuracy. Augmenting by taking a crop of random size (`RandomSizeCenterCrop`), a proxy for scale-invariance of the center object, performs on par with `FixedSizeRandomCrop`, a proxy to invariance to translation, and both bring a substantial improvement compared to no augmentation ( $67.9\%$ vs. $63.5\%$). However, neither is sufficient to fully recapture the performance of RRC. To further test that RRC amounts to translation and scale, we augment the image by translating it by at most 30% in width and height followed by taking a center crop of random size (denoted `T.(30%) + RandomSizeCenterCrop`). This method improves Top-1 accuracy over `RandomSizeCenterCrop` and `FixedSizeRandomCrop`, and almost match RRC but with a gap of $0.75\%$, showing that scaling and translating interact in a precise manner in RRC that is difficult to reproduce with both transformations applied iteratively.

**RCC's performance is driven by translation.** `FixedSizeRandomCrop` impacts translation, but its behavior is contrived, for example an image in the top left corner will never be in the bottom left corner of a crop. Thus to further disambiguate the role of translation versus scale, we also experiment using `T.(30%)` only: we resize the image to 256 on the shorter dimension, apply the random translation of at most 30% and take a center crop of size $224 \times 224$. The gain in performance from `T.(30%)` compared to no augmentation is surprising given the (approximate) translation invariance built in to convolutional architectures such as ResNets. Furthermore, `T.(30%)` performs almost as well as `T.(30%) + RandomSizeCenterCrop`, which has scale augmentation as well. Thus, adding the change in scale to the translation does not further improve performance, which was not the case when comparing `FixedSizeRandomCrop` and `RandomSizeCenterCrop`. We compare the validation images that become correctly classified (compared to no augmentation) when using `FixedSizeRandomCrop`, `T.(30%)` and `T.(30%) + RandomSizeCenterCrop` in Appendix B but no clear pattern emerged.

## 2.2 Trade-off between variance and magnitude of augmentation

What role does the distribution over augmentations in `RandomResizedCrop` play? Default values for the distribution over $s$ are $s_- = 0.08, s_+ = 1$. Thus the scale factor can increase the size of an object in the crop up to to $1/s = 1/0.08 \approx 12.5$ times larger. Does only the range of augmentation magnitudes matter? What if we change the shape of this distribution, for the same range of values?

To test this, we modified `RandomResizedCrop` to use a Beta distribution $B(\alpha, \beta)$ over the standard interval for $s$ ($[0.08, 1]$). Fixing $\alpha = 1$, we vary $\beta$, which changes the distribution shape. Setting $\beta = 1$ corresponds to a uniform distribution $U(0.08, 1)$, while smaller values of $\beta$ lead to heavily sampling values of $s$ near 1 (and vice versa; see Appendix Fig. A1 for visualizations). Table 1b shows that just varying the shape of the distribution results in a $6\%$ drop in performance. To explain this drop, we examine the discrepancy between average apparent object sizes during training and evaluation as in Touvron et al. [33]. We note smaller values of $\beta$ reduce the discrepancy between image sizes during training and evaluation (see Appendix C for the mean of $B(\alpha, \beta)$). However, very small values of $\beta$ (e.g., 0.1) also decrease performance, as they do not encourage scale invariance by sampling $s$ near 1 (no scale change) and $1/s$ has very little variance. Thus, we observe a trade-off between variability to induce invariance and consistency between training and evaluation object sizes.

## 3 Invariance across architectures and augmentations

So far, we have measured the impact of decomposed augmentations on model performance, but how do these augmentations impact invariance to these and other transformations? To what extent do other elements, such as architectural inductive bias and learning contribute to these invariances? And finally, how do these invariances differ across categories of transformations? In this section, we

address these questions by defining a metric for invariance and evaluate this metric for a number of transformations across combinations of architectures, augmentations, and training.

## 3.1 Measuring invariance

A model $f$ is considered invariant to a transformation with a specific magnitude $T_\theta$ if applying $T_\theta$ leaves the output unchanged. We choose to measure invariance by measuring the cosine distance, $d$ between the embeddings of a sample $x$ and its transformed version, relative to a baseline, $b$:

$$\text{Inv}_{T_\theta}(f(x)) = \frac{b - d(f(x), f(T_\theta(x)))}{b} \tag{1}$$

where $f$ generates the embedding up from the penultimate layer of a ResNet18 or a vision transformer trained on ImageNet [14, 36], see Appendix D for training details. The baseline, $b$, is the embedding distance across two randomly selected samples, $b = d(f(x_i), f(T_\theta(x_j)))$, to account for the effect different transformations (and magnitudes) may have on the distribution of embeddings. The closer $\text{Inv}_{T_\theta}$ to 1, the more invariance to $T_\theta$. We report the distribution of $\text{Inv}_{T_\theta}$ across pairs.

To measure how invariance changes as transformations intensify, we report invariance as a function of transformation magnitude, with magnitude 0 meaning no transformation and magnitude 9 corresponding to a large transformation (see Appendix Fig. A2 for examples). We expect invariance to decrease as transformation magnitude increases for the majority of settings.

## 3.2 Invariance to geometric and appearance transformations

To understand the extent and source of invariance, we measured invariance to common appearance and geometric transformations for both ResNet18 and ViT models. Appearance transformations, such as changes in brightness, alter color or illumination, while geometric transformations such as scaling, translation, and rotation alter the spatial arrangement of pixels.

First, we found that models indeed featured invariance to a number of common transformations, including translation and scale (Fig. 2). Examining the impact of architecture, we observed that for translation, both ResNet18 and ViT models learned to be invariant, with ViT models consistently achieving slightly higher translation invariance than ResNet18. Surprisingly, untrained ViT models also featured stronger invariance to translation when compared to untrained ResNet18 (Fig. 2a, b, compare orange with light blue), suggesting that despite the convolutional inductive bias present in ResNet18, translation invariance is more prominent in ViT models.

We next examined the impact of training on invariance. While training consistently increased invariance to translation and to appearance based transformations (Fig. 2a,b and additional figures in Appendix Fig. A3), training resulted in no change in average invariance to scale (zooming-in) but reduced the variability in this invariance (Fig. 2). In contrast, training *reduced* invariance to rotation and shear (Fig. 2d and A3). Finally, we examined the role of augmentation in learning invariance. Consistent with our finding in Sec. 2.1, we found standard augmentation improves translation invariance (albeit only slightly; compare dark and medium blue in Fig. 2a,b). Surprisingly, it barely increases scale invariance, and had equivocal effects on other transformations. Together, these results suggest that the data itself is the major factor influencing learned invariance rather than the choice of architecture or the specific augmentations used.

We have shown that ResNets present less invariance to translation than vision transformers, despite the inductive bias of the convolutional architecture. As such, what role does the architectural inductive bias in ResNets play? To answer this, we measure equivariance to assess whether models encode predictable responses to transformations. As described in Section 1, equivariance is a more general property than invariance: Invariance is a specific case of equivariance and equivariance does not necessarily imply invariance (though invariance does imply equivariance). We evaluate equivariance by examining the alignment of embedding responses to transformations. We find while an untrained ResNet18 is not invariant to translation, it is equivariant to translation, highlighting the architectural inductive bias of CNNs to translation (see Appendix D for detailed results).

Nevertheless, we observed that trained ViT and ResNet18 are both able to learn invariance to both geometric and appearance transformations, regardless of their particular inductive biases. However, it remains unclear how these learned invariances relate to the factors of variation present in ImageNet. In the next section, we attempt to answer this question.

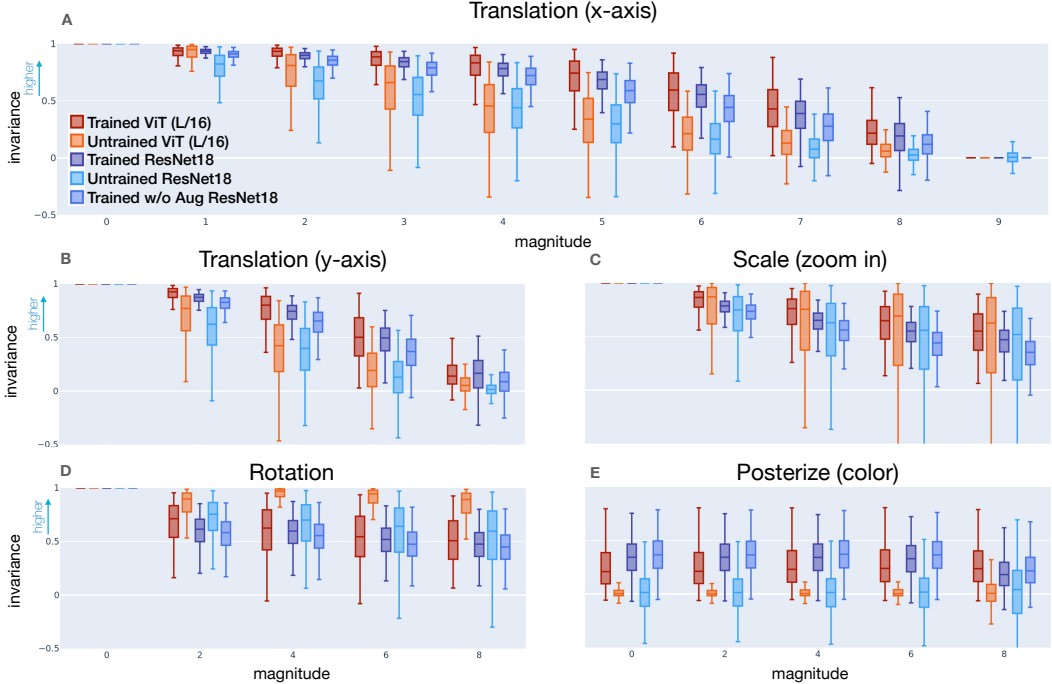

Figure 2: **Comparing sources of invariance**. We compare invariance across trained and untrained ResNet18 and ViT to isolate the effect of architectural bias, training, and data augmentation on invariance. Standard augmentations used for ResNet18 and ViT are RRC+horizontal flips. Values are for training pairs, though we found similar trends on validation pairs.

## 4 Characterizing factors of variation with similarity search

In the previous section, we assessed the extent and source of invariance to transformations for ResNet18 and ViT. Among learning, data augmentation, and architectural inductive biases, we identified learning from data as the predominant source of invariance. Consequently, here we investigate which aspects of the data's variation drive models' learned invariances. We characterize the variation in ImageNet in terms of transformations and relate the variation to the models' learned invariances. To do so, we design a metric comparing how well each transformation allows us to travel from one image to another image with the same label, thus capturing a factor of variation.

In the previous section, we assessed whether ResNet18 and ViT are invariant to a set of transformations, and explored to what extent learning, data augmentation and architectural inductive biases impact this invariance. But do learned invariances correspond to the transformations that actually affect factors of variation in the data? What are the transformations that explain variations in ImageNet? In this section, we design a metric to answer these questions by comparing how well each transformation allows us to travel from one image to another image with the same label, thus capturing a factor of variation. Finally, we relate our findings to models' invariances from the previous section.

Characterizing the factors of variation present in natural images is challenging since we don't have access to a generative model of these images. Here, we introduce a metric to determine these transformations based on a simple idea: factors of variation in the data should be able to explain the differences between images of the same class. For example, suppose one of the primary factors of variation in a dataset of animals is pose. In this case, by modifying the pose of one image of a dog, we should be able to match another image of a dog with a different pose. Concretely, we measure the change in similarity a transformation brings to image pairs. For an image pair $(x_1, x_2)$ from the same class and a transformation $T_\theta$ we measure the percent similarity change as

$$\text{SimChange}_{T_\theta} = \frac{sim(f(x_1), f(T_\theta(x_2))) - sim(f(x_1), f(x_2))}{sim(f(x_1), f(x_2))} \quad (2)$$

where $sim$ measures cosine similarity (see Appendix E). To control for any effect from data augmentation, we take $f$ to be a ResNet18 model up to the penultimate layer trained without data

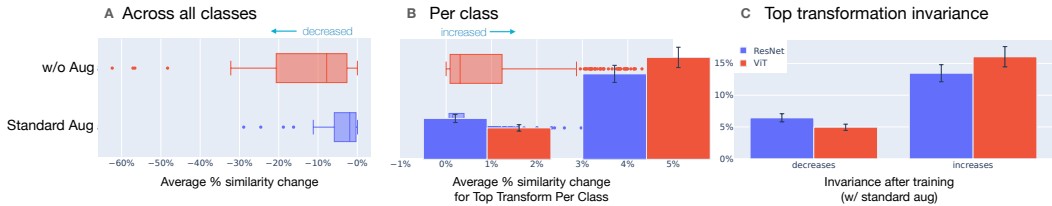

Figure 3: **Factors of variation are class-specific and have higher invariance after training**. Fig A. Compares the average percent change in similarity of pairs across all classes. No single geometric or appearance transformation increases pair similarity across all classes. In contrast, Fig. B shows per class transformations consistently increase similarity. Fig C. shows training increases invariance for transformations more likely to appear among the top 5 transformations per class.

augmentation[3] and report values on training image pairs. A higher similarity $\text{SimChange}_{T_\theta}$ among pairs implies a stronger correspondence between the transformation and factors of variation. We select a relevant pool of transformations using AutoAugment, an automated augmentation search procedure across several image datasets [11]. The set of transformations encompasses geometric and appearance transformations of varying magnitudes[4]. For each $T_\theta$ we report $\text{SimChange}_{T_\theta}$ averaged on image pairs.

This metric has both advantages and disadvantages. Its primary advantage is that it can be measured on realistic image datasets such as ImageNet without requiring access to a ground-truth generative model, as is often used in synthetic datasets. It also uses a pool of possible transformations which are realizable with standard data augmentation techniques, and encompasses both geometric and appearance based transformations. However, because this metric does not exploit information about the generative process, it has limited ability to capture realistic transformations that occur in the abstract space describing semantic image content. Furthermore, it is difficult to make conclusions based on absolute metrics; as such, we use relative comparisons.

## 4.1 Can image pairs across ImageNet be described by a consistent set of transformations?

Underlying the common practice of data augmentation is the assumption that images ought to vary consistently across a dataset. Throughout training, samples are augmented with the same set of transformations, albeit with differing magnitudes. To test this assumption we ask whether we can explain the variation among image pairs with the same set of transformations. If so, we expect to find a set of transformations which consistently increases the similarity of pairs. In Fig. 3A we show the distribution of average similarity changes across each transformation as measured by Equation 2. We observe that no single transformation consistently increases average similarity of image pairs across all classes, including geometric and appearance transformations. We find the same pattern holds whether a ResNet18 is trained with or without standard augmentations (RRC + horizontal flips). This result suggests that although standard approaches to data augmentation apply the same transformation distribution to all classes, no single transformation consistently improves similarity (see Appendix E). Could we instead consistently increase similarity among image pairs by a combination of transformations?

**Combinations of transformations do not consistently increase similarity.** We repeat our analysis of transformations' effect on image pair similarity using sub-policies, which combine two transformations of varying magnitudes. We found that while sub-policies can help or hurt by a larger margin as they apply multiple transformations, no sub-policy increases average similarity across all classes (Fig. A4).

**Does using combinations of transformations increase similarity?** To answer this, we repeat the analysis using sub-policies, which combine two transformations of varying magnitudes. We found that while sub-policies can help or hurt by a larger margin as they apply multiple transformations, no sub-policy increases average similarity across all classes (Fig. A4).

---

[3]Note that if $f$ is fully invariant to $T_\theta$ then $\text{SimChange}_{T_\theta}$ will be zero. Thus we also chose $f$ trained w/o augmentation to reduce translation invariance and confirmed full invariance is not achieved in Sec. 3.

[4]See Appendix A2 for full list of possible transformations.

Counter to the common practice of data augmentation, this result suggests the same set of transformations does not consistently explain the variation among image pairs. What effect then, if any, does augmenting with the same set of standard transformations have on image similarity?

**Training augmentations dampen pair similarity changes.** Similar to our earlier results, we observed that training with augmentations induce more invariance relative to training without augmentations. Though all transformations consistently decrease similarity for models trained with and without augmentation, models trained with augmentation exhibited both a smaller decrease in similarity and lower variance (Fig. 3A). Training with augmentation also reduced the decrease in similarity for transformations beyond simply scale and translation (Appendix Fig. A5), suggesting that these augmentations impact the response even to transformations not used during training.

## 4.2  Are factors of variation specific to each class?

In Sec. 4.1, we showed no transformation (or sub-policy) consistently increased similarity across classes. However, the factors of variation and consequently, the optimal transformation, may be different for different classes, especially those which are highly different. Can we instead consistently increase similarity if we allow flexibility for transformations to be class specific?

To test this hypothesis, we examined the top transformation for each class. In contrast to the global result, we found the top transformation per class consistently increased the average similarity for all classes (Fig. 3B). Notably, the top transformation per class increased similarity by $3.36 \pm 0.9\%$ (mean $\pm$ SEM) compared to effectively no change ($0.020\% \pm 0.021\%$) for the top transformation across all classes. Similar to Hauberg et al. [21] which learn per class transformations for data augmentation that boost classification performance on MNIST [24], we applied per class data augmentation variants on ImageNet, but observed no significant classification performance boost. We leave further applications of per class augmentation to future work.

**Data augmentation is not beneficial for all classes.** Since the optimal transformation differs across classes, do standard augmentations benefit all classes or only some classes? To test this, we examined the impact of RRC on the performance of individual ImageNet classes. Interestingly, we found that on average[5] $12.3 \pm 0.21\%$ have a lower performance when using RRC. Critically, some classes are consistently hurt by the use of RRC with a difference in top-1 accuracy as high as $22\%$. We systematically study these classes in Appendix D.1, but no clear pattern emerged.

## 4.3  Appearance transformations are more prevalent

Data augmentation and most of the literature on invariant models often rely only on geometric transformations such as translations, rotations, and scaling. However, it is not clear whether or not the factors of variation in natural images are primarily geometric. If this is the case, we would expect the top transformations from our similarity search to be geometric rather than appearance-based. In contrast, we found that appearance transformations accounted for more variation in ImageNet than geometric transformations, consistent with recent work [11]. Of the top transforms, $78\%$ were appearance based compared to only $22\%$ geometric. We confirmed this difference is not due to a sampling bias by ensuring an approximately even split between geometric and appearance transformations. In fact, if we isolate geometric transformations, we find for $64\%$ of classes the top transformation is the identity, suggesting geometric transformations are worse when applied to an entire class than no transformation at all. We find a similar pattern among the top transforms per class for ResNet18 trained with standard augmentations: for more than $98.4\%$ of classes the top transformation alters appearance not geometry. In Appendix E.1 we also examine local variation in foregrounds to translation and scale.

## 4.4  Training increases invariance for factors of variation

In Sec. 3, we showed that training increases invariance to a number of transformations, independent of architectural bias and augmentation and in Sec. 4.1, we used similarity search to characterize the transformations present in natural images. However, do the invariances learned over training correspond to the factors of variation present in natural images?

---

[5]Computed over 25 pairwise comparisons of 5 runs with both augmentations.

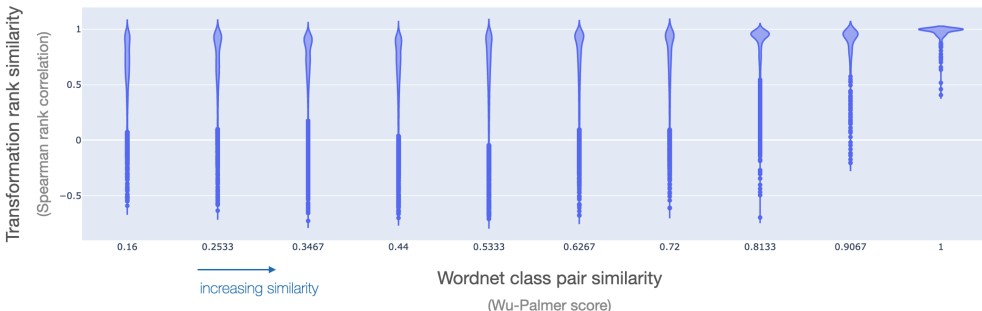

Figure 4: **Similar classes share factors of variation**. Wordnet class pair similarity versus transformation ranking. Violin plots show the distribution of transformation Spearman's rank correlation as class pair similarity increases. We see similar classes rank transformations similarly.

To test this, we asked whether the transformations to which networks learn to be invariance correspond to the same transformations which are highly ranked in similarity search. We found transformations that exhibit increased invariance over training were substantially more likely to be ranked in the top 5 transformations per class compared to transformations which exhibited decreased or minimal change in invariance over training (Fig. 3C). This result demonstrates training increases invariance to factors of variation present in real data, regardless of whether there is an inductive bias or data augmentation is specifically designed to encourage invariance.

### 4.5   Characterizing factors of variation across classes

We have shown that there exist factors of variation which consistently increase similarity for a given class, but it remains unclear why a particular factor might impact a particular class. Here, we investigate whether related classes feature related factors of variation.

One prominent pattern which emerged was among textile-like classes such as velvet, wool, handkerchief, and envelope. When considering single transformations, rotation is the top transformation or rotation plus an appearance transformations (such as color or posterize) for sub-policies (see Appendix E.2 for a full list). The relationship between rotation and textiles makes intuitive sense: fabrics generally don't have a canonical orientation and can appear in many different colors.

To test this systematically, we measured the pairwise class similarity using Wordnet [16] and compared it to the similarity between the top transforms for each class. We computed class similarities using the most specific common ancestor in the Wordnet tree against the Spearman rank of transformation types. We found that while dissimilar classes often have similar transformations, similar classes consistently exhibit more similar transformations (Fig. 4; Appendix E.3).

## 5   Related work

**Data augmentation approaches.**   Standard data augmentations often amount to taking a crop of random size and location, horizontal flipping and color jittering [22, 32, 33]. In self-supervised learning, [7] boost performance by using multiple crops of the same image. Hauberg et al. [21] learn per class augmentation and improve performance on the MNIST dataset [24]. AutoAugment is a reinforcement learning-based technique that discovers data augmentations that most aid downstream performance [11]. Antoniou et al. [3] train a Generative Adversarial Networks (GANs [18]) based model to generate new training samples. van der Wilk et al. [34] follow a Bayesian approach and integrate invariance into the prior. Recent works aim to automatically discover symmetries in a dataset [20], and enforce equivariance or invariance to these [6, 12, 39]. While these methods are promising, they have mostly been applied to synthetic datasets or augmented versions of real datasets. Their application to a real dataset such as ImageNet is not straightforward: we tried applying the Augerino model [6] to ImageNet, we found it was struggling to discover effective augmentations composed of multiple transformations (see Appendix G).

**Consistency of neural architectures.**   Zhang [38] show that invariance to translation is lost in neural networks, and propose a solution based on anti-aliasing by low-pass filtering. Kauderer-

Abrams [23] studies the source of CNNs translation invariance of CNNs on a translated MNIST dataset [24] using translation-sensitivity maps based on Euclidean distance of embeddings, and find that data augmentation has a bigger effect on translation invariance than architectural choices. Very recently, Myburgh et al. [27] replace Euclidean distance with cosine similarity, and find that fully connected layers contribute more to translation invariance than convolutional ones. Lenc and Vedaldi [25] study the equivariance, invariance and equivalence of different convolutional architectures with respect to geometric transformations. Here, we study invariance on a larger set of transformations with varying magnitudes and compare ResNet18 and ViT. Engstrom et al. [15], Touvron et al. [33] explore the specificities of standard data augmentations, and Engstrom et al. [15] found that a model augmented at training with rotations and translations still fails on augmented test images. We differ from these works by grounding invariance into the natural factors of variation of the data, which we try to characterize. To the best of our knowledge, the links between invariance and the data factors of variation has yet not been studied on a large-scale real images dataset.

## 6   Discussion

In this work, we explored the source of invariance in both convolutional and vision transformer architectures trained for classification on ImageNet, and how these invariances relate to the factors of variation of ImageNet. We compared the impact of data augmentation, architectural bias and the data itself on the models' invariances. We observed that `RandomResizedCrop` relies on a precise combination of translation and scale that is difficult to reproduce and that, surprisingly, augmenting with translation recaptures most of the improvement despite the (approximate) invariance to translation built in to convolutional architectures. By analyzing the source of invariance in ResNet18 and ViT, we demonstrated that invariance generally stems from the data itself rather than from architectural bias, and is only slightly increased by data augmentation.

Finally, we connected the models' learned invariance to the factors of variation present in the data. We characterized variation in ImageNet by examining pair similarity in response to transformations, finding that transformations which explain variation in ImageNet are class-specific, more appearance-based, and align with the invariances' learned during training.

**Limitations and future work**   We provide an analysis of the invariant properties of models using a specific set of metrics based on model performance and similarities of input embeddings. Using these, some of our conclusions are shared with existing works, but a different set of metrics could potentially bring more insights on our conclusions. Additionally, we only experiment on ImageNet, but it would be interesting to perform the same type of analysis on a wider range of datasets and data types. Does a handful of transformations describe the variations of most standard image datasets? Our study sheds light on ImageNet factors of variation but some conclusions remain obscure, such as the role of scaling. This emphasizes the difficulty of performing a systematic study of real datasets.

Finally, our findings spark further exploration. Could tailoring augmentations per class or introducing appearance-based augmentations improve performance?

**Potential negative societal impacts.**   While our work is concerned with robustness of vision models which can have various societal impacts, our work is primarily analytical and we do not propose a new model. Hence, we do not foresee any potential negative societal impacts of our findings. Our study does emphasize the importance of dataset construction, as the predominant source of invariance, relative to other modelling considerations. Consequently, our work encourages researchers to also consider dataset construction as an important aspect of vision models' societal impact.

**Acknowledgements**   We would like to thank David Lopez-Paz, Armand Joulin, Yann Olivier, Kamalika Chaudhuri, Aaron Aadock, Pascal Vincent, Michael Alcorn, and Nicolas Ballas for helpful discussions and feedback.

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
