## Appendix A    Training details

**Training regular Resnet-18**    The experiments of Sec. 2.1 are conducted using 1 seed to cross-validate between 3 learning rates $(0.01, 0.1, 0.5)$ and 3 weight decay parameter $0.01, 0.001, 0.0001$. Models are trained with Stochastic Gradient Descent with momentum equal to 0.9 [30] on all parameters. We use a learning rate annealing scheme, decreasing the learning rate by a factor of 0.1 every 30 epochs. We train all models for 150 epochs. Then, we select the best learning rate and weight decay for each method and run 5 different seeds to report mean and standard deviation. We use the validation set of ImageNet to perform cross-validation and report performance on it. Our code is a a modification of the pytorch example found in `https://github.com/pytorch/examples/tree/master/imagenet`.

Note that we also tried one seed with cross-validation of hyper-parameters of `T.(50%) + RandomSizeCenterCrop`, i.e. with $50\%$ translation, this gives poorer performances than $30\%$ translation (top-1 accuracy $\approx 67.5\%$).

Code to reproduce experiments is available at `https://github.com/facebookresearch/grounding-inductive-biases`.

**Training the Augerino model**    In section G we train the Augerino method on top of the Resnet-18 architecture. We employ Augerino on top of applying the `FixedSizeCenterCrop` pre-processing, in order to not induce any invariance by data augmentations. The experiments reported in section G are conducted using 5 seeds to cross-validate between 7 regularization values $\lambda$ $(0.01, 0.1, 0.2, 0.4, 0.6, 0.8, 1)$. We use the best learning rate and weight decay values of the Resnet-18 trained with `FixedSizeCenterCrop` (learning rate $0.1$ and weight decay $0.0001$). The parameters of the distribution bounds, specific to Augerino, are trained with a learning rate of $0.01$ and no weight decay as in the original Augerino code (`https://github.com/g-benton/learning-invariances`). Models are trained with Stochastic Gradient Descent with momentum equal to 0.9 [30] on all parameters. We use a learning rate annealing scheme, decreasing both learning rates of the Resnet-18 and the Augerino bounds parameters by a factor of 0.1 every 30 epochs. We train all models for 150 epochs. During training, 1 copy of the image transformed with the sampled transformation values is used, and during validation and test, 4 transformed versions of the image are used, as in the original Augerino code. We use the validation set of ImageNet to perform cross-validation and report performance on it.

**Total amount of compute, type of resources used**    Our main code runs with the following configuration: pytorch 1.8.1+cu111, torchvision 0.9.1+cu111, python 3.9.4. Full list of packages used is released with our code. We use `DistributedDataParallel` and ran the experiments on 4 GPUs of 480GB memory (NVIDIA GPUs of types P100 and V100) and 20 CPUs, on an internal cluster. With this setting, training a ResNet-18 on ImageNet with `RandomResizedCrop` takes approximately 6 mins, while for other augmentations (e.g.`T.(30%) + RandomSizeCenterCrop`) it can take up to approximately 20 mins depending on the type of GPU used.

For the experiments of Sec. G we use pytorch 1.7.1+cu110, torchvision 0.8.2+cu110, python 3.9.2 and torchdiffeq 0.2.1 as the Augerino original code relies on torchdiffeq and we could not run torchdiffeq with pytorch 1.8.1 (known issue, see `https://github.com/rtqichen/torchdiffeq/issues/152`).

## Appendix B    Comparing the samples helped by translation and/or scaling

In Section 2.1, we find that `T.(30%)` performs on par with `T.(30%) + RandomSizeCenterCrop`, and both outperform `FixedSizeRandomCrop`. To further study this, we compute the lists of samples that are incorrectly classified by no augmentation but correctly classified by these methods, for each of the three methods. As we trained 5 seeds per methods, we have 25 lists of "helped samples" for each method. We then compare methods using the intersection-over-union (IoU) of their respective lists. For each method, the IoUs of each method's lists with itself are:

- `T.(30%)`/`T.(30%)`: $0.269 \pm 0.007$

- `T.(30%) + RandomSizeCenterCrop/T.(30%) + RandomSizeCenterCrop`: $0.293 \pm 0.007$

- `FixedSizeRandomCrop/FixedSizeRandomCrop`: $0.242 \pm 0.007$

while the cross-methods lists IoU are:

- `FixedSizeRandomCrop/T.(30%) + RandomSizeCenterCrop`: $0.220 \pm 0.004$

- `FixedSizeRandomCrop/T.(30%)`: $0.224 \pm 0.005$

- `T.(30%)+RandomSizeCenterCrop/T.(30%)`: $0.242 \pm 0.005$

Thus, we do not see a pattern of consistency neither in the intra-methods or cross-methods IoUs.

## Appendix C   Varying the distribution over the scale

For the experiment in Sec. 2.2, we used the best learning rate and weight decay (0.1 and 0.0001) found by cross-validation for `RandomResizedCrop`. We run 5 training seeds of each distribution.

The expected value of the non-standard Beta distribution for $\alpha = 1$ is $\frac{1}{1 + \beta}(1 - 0.08) + 0.08$. As explained in Touvron et al. [33], using a random scale induces a discrepancy between the average objects apparent sizes during training and evaluation: their ratio is proportional to the expected value of $s$. Thus, the smaller $\beta$ the smaller the expected value of $s$, which reduces this discrepancy.

However, too small values of $\beta$ (e.g. $\beta = 0.1$) also have smaller performance. Recall that `RandomResizedCrop` scales an object in the selected crop by a factor proportional to $1/s$. While we don't have a closed form expression for $\mathbb{V}[1/s]$, the variance of $1/s$ (inverse of the non-standard Beta over $s$), we computed estimated values for $\mathbb{V}[1/s]$ using $500000$ samples drawn from $s$. [6] Table A1

| $\beta$ | $\mathbb{V}[1/s]$ |
|---|---|
| 0.1 | 0.823 |
| 0.5 | 3.193 |
| 1 ($\sim$ RRC) | 4.962 |
| 2 | 6.809 |
| 3 | 7.626 |
| 10 | 7.317 |

Table A1: Variance of $1/s$.

shows that the variance for $\beta = 0.1$ is $\approx 3$ and 5 times smaller than the variance for $\beta = 0.5$ and $\beta = 1$ respectively. Thus, while the three values of $\beta$ give distributions that peak close to $1/s = 1$, the value $\beta = 0.1$ gives a smaller variance thus less encourage scale invariance. This in our view explains the poorer performance of $\beta = 0.1$.

## Appendix D   Invariance and Equivariance

**Experimental details**   We use a pretrained ViT (L/16) and untrained ViT from [36] using the 'forward features' method to generate embeddings, see Timm's documentation for details `https://rwightman.github.io/pytorch-image-models/feature_extraction/`. The trained ViT achieves $80\%$ Top 1 accuracy on ImagNet. For ResNet18, we use the pretrained ResNet18 available from PyTorch `https://pytorch.org/hub/pytorch_vision_resnet/`. The trained ResNet18 achieves $70\%$ Top 1 accuracy. For all our measures, we control for the difference in test-set versus train-set sizes by limiting the total number of embedding comparisons to $10k$ pairs.

---

[6]We also checked the estimated values were sensible obtained by computing the probability distribution function (pdf) of $1/s$ from the pdf of $s$ via the change of variable formula, and compute $\mathbb{V}[1/s] = \mathbb{E}[1/s^2] - \mathbb{E}[1/s]^2$ using Wolfram Alpha [37].

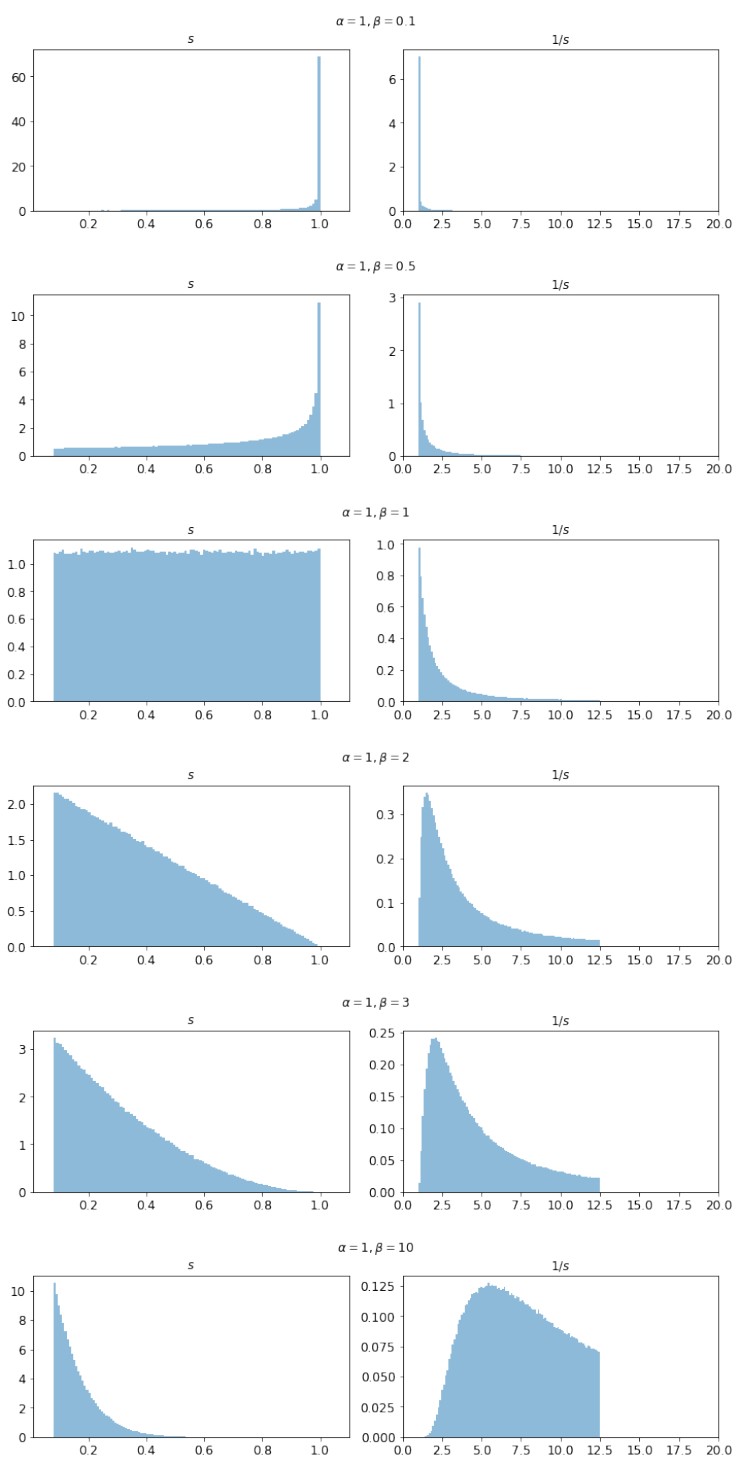

Figure A1: Effect of the value $\beta$ over the distribution of $s$ and $1/s$.

**Equivariance**   We also explore whether models are equivariant to transformations. A model is said to be equivariant if it responds predictably to the given transformation. Formally, a model, $f$, is equivariant to a transformation $T_\theta$ of an input $x$ if the model's output transforms in a corresponding manner via an output transformation $T'_\theta$, i.e. $T'_\theta(f(x)) = f(T_\theta(x))$ for any $x$.

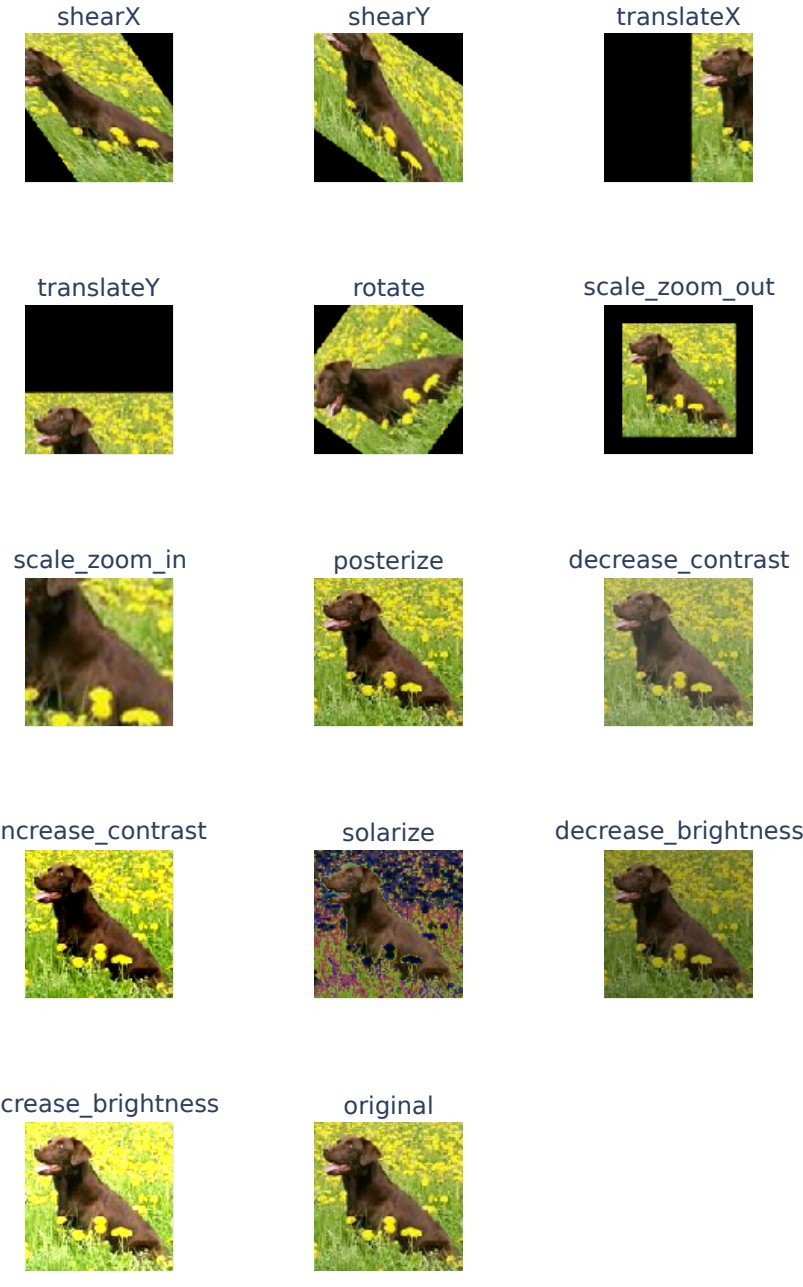

Figure A2: Illustrates the effect of each transformation. The image is from Wikimedia Commons `https://commons.wikimedia.org/wiki/File:Labrador_Chocolate.jpg` under Creative Commons Attribution-Share Alike 3.0 Unported license.

To disambiguate invariance from equivariance, we measure equivariance by examining whether embeddings respond predictably to a given transformation. To do so, we measure alignment among embedding differences, by first producing embedding differences

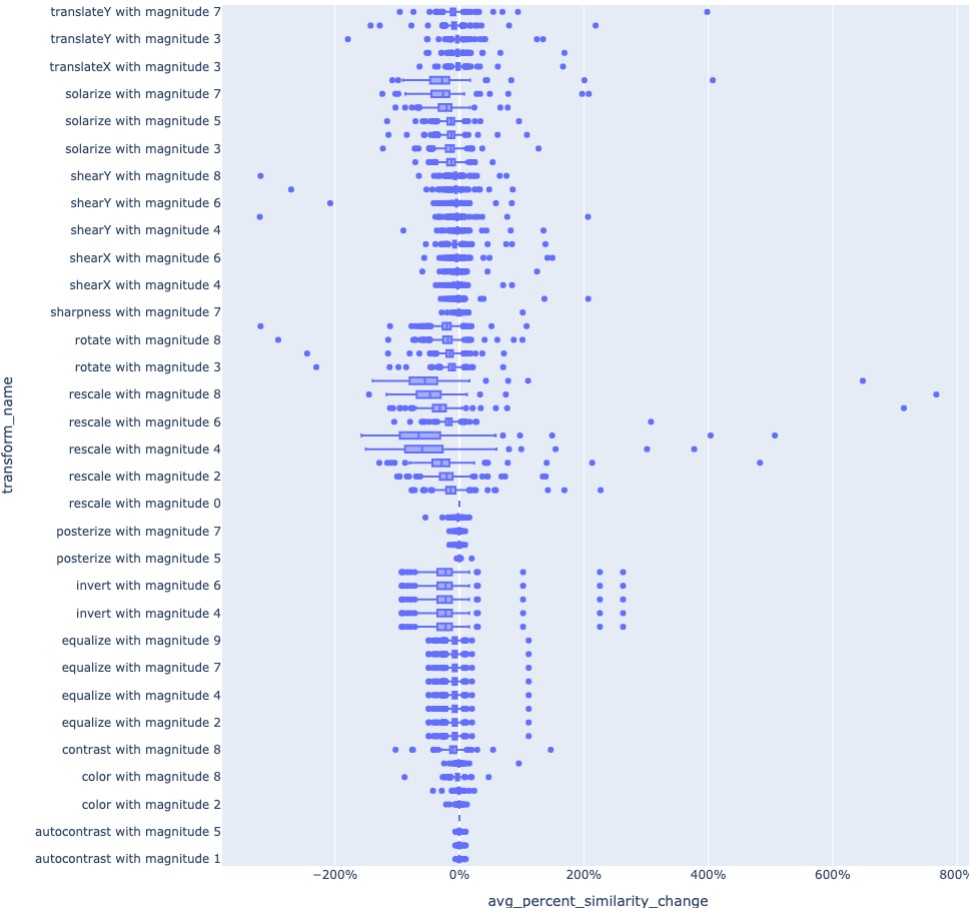

Figure A3: Similarity Changes by transformation across all classes measured on ImageNet training samples (we find a similar pattern on the validation set). We see no transformation similarity change distribution are around or below zero.

$$d_i = f(x_i) - f(T_\theta(x_i))$$

then measuring pairwise alignment of the embedding differences via cosine similarity. We compare, $sim(d_i, d_j)$ against a baseline B where we shuffle the rows in each column independently. We report $sim(b_i, b_j) - sim(d_i, d_j)$, where $b$ are elements from the baseline. A higher value implies higher equivariance, with 0 indicating no equivariance above the baseline. In Fig. A10, we find equivariance to translation for untrained ResNet18 that is absent for ViT, highlighting the architectural inductive bias of CNNs to translation. Although for some magnitudes we also observe equivariance to zooming out, we note this is likely due to zooming out introduce padding rather than true equivariance to scale changes. We also observe equivariance to appearance transformations for ResNet18 such as posterize that are also absent from ViT.

### D.1 Classes consistently hurt by `RandomResizedCrop`

We compare the per-class top-1 accuracy when using `RandomResizedCrop` augmentation training versus FixedSized CenterCrop augmentation (which is the no augmentation). On average, over 25 pairwise comparisons of 5 runs with both augmentations, $12.3\% \pm 0.21\%$ of classes are hurt by the use of `RandomResizedCrop`. We list in Table A2 the classes that are hurt in more than $70\%$ of the 25 comparisons, and the average amount of decrease in Top-1 accuracy when incurred. We do not see a pattern in these classes when reading their labels. We confirm the lack of pattern by

| Class | % of comparisons where hurt | Loss in Top-1 accuracy (%) |
|---|---|---|
| cassette player | 100.0 | 22.00 |
| maillot | 100.0 | 21.20 |
| palace | 100.0 | 4.40 |
| academic gown | 92.0 | 13.57 |
| missile | 88.0 | 11.82 |
| mashed potato | 88.0 | 9.09 |
| digital watch | 84.0 | 7.52 |
| barn spider | 80.0 | 7.70 |
| Indian elephant | 80.0 | 7.20 |
| miniskirt | 80.0 | 8.00 |
| pier | 80.0 | 5.30 |
| wool | 80.0 | 6.80 |
| ear | 80.0 | 10.60 |
| brain coral | 72.0 | 3.89 |
| crate | 72.0 | 4.33 |
| fountain pen | 72.0 | 5.22 |
| space bar | 72.0 | 6.33 |

Table A2: Validation classes hurt by `RandomResizedCrop`

computing the similarities of the classes listed in Table A2 using the most specific common ancestor in the Wordnet [16] tree. The similarity of the classes that are consistently hurt (17 classes) is $0.42 \pm 0.013$ ($0.49 \pm 0.019$ if we include a class similarity with itself), while the similarity between the classes that are consistently hurt and the ones that are consistently helped ($> 70\%$ of comparisons) is $0.406 \pm 0.0014$, and between classes consistently hurt and everything else (neither helped or hurt) is $0.403 \pm 0.002$.

## Appendix E    Similarity Search

**Experimental details**    We use the same trained models using standard augmentations as we did for equivariance. For the no augmentation ResNet18, we use the training procedure outlined in Appendix A. We sample $1k$ pairs from each class and compute SimChange$_{T_\theta}$ for each pair. We pool transformations from subpolicies discovered by AutoAugment on ImageNet, SVHN, and additional rescaling for zooming in and out. We extend the implementation of AutoAugment provided in the DeepVoltaire library `https://github.com/DeepVoltaire/AutoAugment`. Note we disregard the learned probabilities from AutoAugment and instead apply each transformation independently for our similarity search analysis. For subpolicies, we apply each transformation in sequence. Transformations include 'equalize', 'solarize', 'shearX', 'invert', 'translateY', 'shearY', 'color', 'rescale', 'autocontrast', 'rotate', 'posterize', 'contrast', 'sharpness', 'translateX' with varying magnitudes. We illustrate the effect of transformations in Fig. A2.

**No transformation consistently increases pair similarity**    In Fig. A3 we show the distribution of similarity change of each transformations over classes. While for some outlier classes, some transformations increase similarity among pairs, distributions are below or near 0 for all transformations. The top single transformation across all classes is posterize, which increases similarity by $0.02 \pm 0.02 (SEM)$, implying no statistically significant increase. In contrast, we find on average 6.5 out of the top 10 transformations *per class* increase similarity by $5.4\% \pm 0.04\% (SEM)$.

**Geometric transformations**    We also examine the distributions for transformations not in the standard augmentations by excluding all translations and rescales. In Fig. A5 we see even if we exclude translation and scale, standard data augmentation drastically decrease the variation of similarity changes even for other transformations not used during training.

### E.1    Measuring local variation in foregrounds

To supplement our analysis of global transformations, we also directly measure local variation in ImageNet using foregrounds extracted from U2Net [31] trained on DUTS [35]. We measure the center coordinates and area of bounding boxes around the foreground object relative to the image

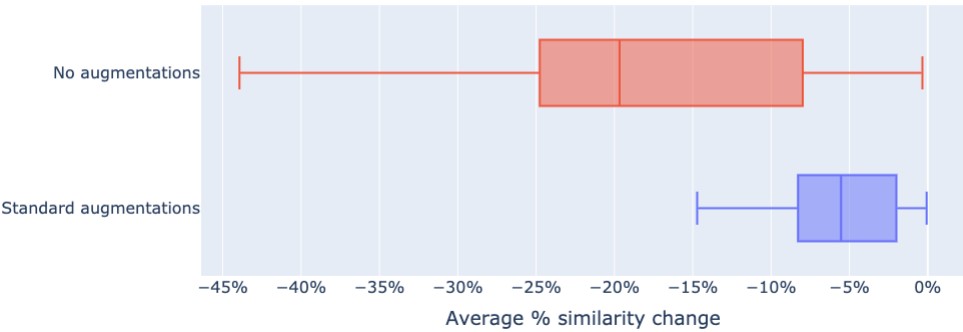

Figure A4: Similarity Search average change across classes for the best subpolicies discovered by AutoAugment on ImageNet. We see no subpolicy consistently increases similarity among pairs.

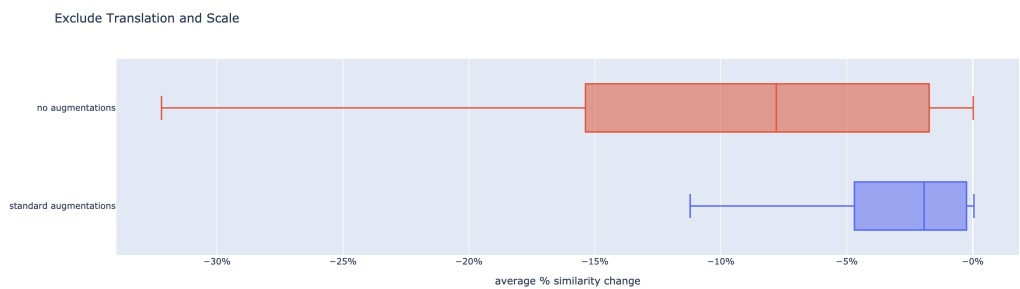

Figure A5: Similarity Search excluding translation and scale which are used during standard augmentation.

frame using a threshold of $0.01$ to determine the bounding box. We measure foreground variation over all training images in ImageNet. We observe there is more variation in scale, which ranges from $31\% - 74\%$ of the image, compared to translation which is centered ($50\% \pm 5\%$ (SEM)).

## E.2 Textiles Weighted Boost

To account for both the size of the similarity increases and the proportion of images increased, we rank transformations by their weighted boost, defined as the average percent boost * proportion of image pairs boosted. We examine the top 10 transformations by weighted boost and find rotate with magnitudes ranging from 3-9 is the top transformation for all top 10 for the ResNet18 trained with standard augmentation. We find for rotation the corresponding classes are velvet, handkerchief, envelop, and wool with velvet appearing 4 times among the top 10. For subpolicies, we find both the ResNet18 trained with or without standard augmentations, rotation and a color transformation with varying magnitudes is the top 10 transformation also corresponding to velvet, wool, handkerchief, and envelope with jigsaw puzzle as an additional class.

## E.3 Wordnet Similarity Search

To study whether similar class have similar factors of variation, we measured class similarity using the WordNet hierarchy. We compute similarity using several methods provided in the NLTK library `https://www.nltk.org/howto/wordnet.html` including Wu-Palmer score, Leacock-Chodorow Similarity, and path similarity—all of which compute similarity by comparing the lowest common ancestor in the WordNet tree. To compute similarity of transformations, we compared the Spearman rank correlation of the top transformation in each class by average percent similarity change as well as proportion of image pairs boosted. We found no significant difference between the two. We compare class WordNet similarity against transformation ranks for all $1k$ ImageNet classes.

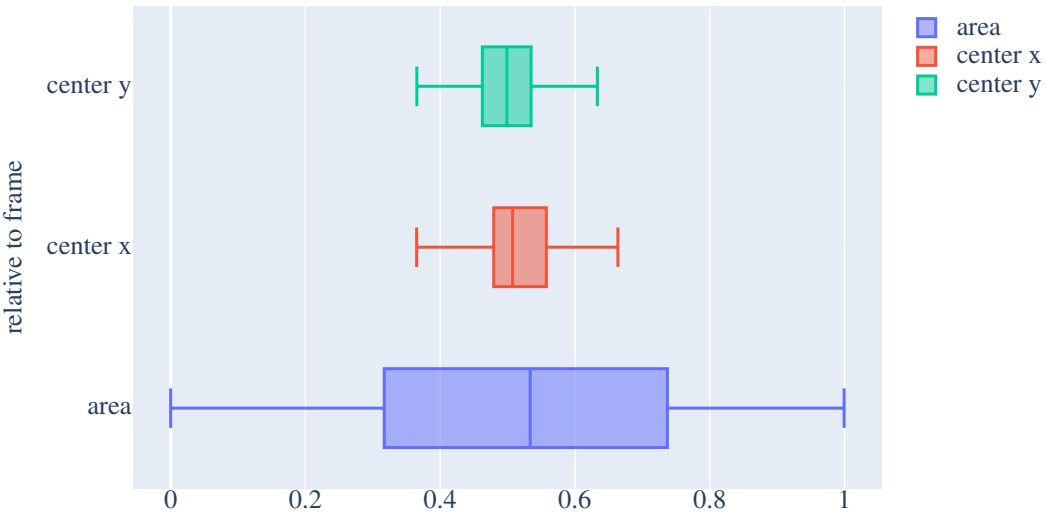

Figure A6: Local variation in foregrounds.

## Appendix F  Additional experiment: How does augmenting validation images affect accuracy?

Standard data augmentation methods are a proxy to implement geometric transformations such as translation and scale, do they actually bring invariance to these transformations? If so, the performance of a model trained with data augmentation should be equal even when validation images are augmented. To answer this question, similar to Engstrom et al. [15], we augment the images during evaluation to test for scale invariance. Indeed, robustness to augmentations is used a generalization metric in Aithal et al. [1].

Specifically, we augment the validation images with the regular validation pre-processing `FixedSixeCenterCrop`, and then augment the images by taking a `RandomSizeCenterCrop` (disabling aspect ratio change). This scales the object in the crop. Using `RandomSizeCenterCrop`, we vary $s_-$, that specifies a lower bound of the uniform distribution $s \sim U(s_-, 1)$. This effectively varies the maximal increase in size potentially applied, which is proportional to $1/s_-$. We compare the models trained with `RandomResizedCrop` and `FixedSizeCenterCrop` (no augmentation).

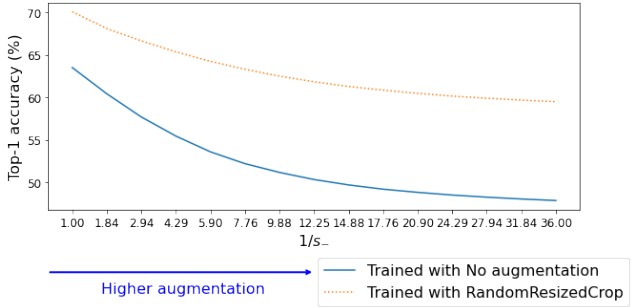

Figure A7: Augmenting the validation set.

Specifically, we select 15 values of increase in size per-axis $v$ uniformly between 1 and 6, and set $s_- = 1/v^2$. Thus we augment the images from no augmentation ($s = 1$) to scaling the image by a factor potentially as large as 36 when $v = 6$. Note that for the value $v = 3.5$, $s_- = 1/v^2 \approx 0.08$, matching the value of `RandomResizedCrop` lower bound on the range of $s$. To match the procedure done at evaluation, we use `FixedSizeCenterCrop`, and then augment the crop. For each model, we evaluate the 5 training seeds of the best hyper-parameters setting, running each augmentation experiment for 5 different test seeds. We compute performance averaged over the test seeds and then report the mean of the 5 training seeds $\pm$ standard error of the mean.

Fig. A7 shows that the accuracy on the validation set decreases as we increase the magnitude of augmentations on validation images. The stronger decrease is for the no augmentation model, while the model trained with `RandomResizedCrop` is more robust. However, while the latter has been trained with augmentations up to $1/s_- \approx 12.5$, its performance already decreases for the smallest values[7]. This suggests it might only present *partial* invariance.

## Appendix G  Additional experiment: Augerino [6] on ImageNet

We have shown that standard data augmentation methods rely on a precise combination of transformations and parameters, and needs to be hand-tuned. To overcome these issues, and potentially discover relevant factors of variation of the data, recent methods have been proposed to automatically discover symmetries that are present in a dataset [6, 12, 39]. We assess the potential of a state-of-the-art model of this type to tackle ImageNet, that is, the Augerino model [6].

### G.1  The Augerino model and our modifications

**Augerino method**   Augerino is a method for automatic discovery of relevant equivariances and invariances from training data only, given a downstream task. Given a neural network $f_w$ parametrized by $w$, Augerino creates $\hat{f}$ a model approximately invariant to a group of transformations $G$ by averaging the outputs over a distribution $\mu_\theta$ over $g \in G$:

$$\hat{f}_w(x) = \mathbb{E}_{g \sim \mu_\theta}[f_w(gx)]. \tag{3}$$

Augerino considers the group of affine transformations in 2D, Aff(2), composed of 6 generators corresponding to translation in $x$, translation in $y$, rotation, uniform scaling, stretching and shearing.[8]. Instead of directly using a distribution over transformations in image space $g$, the distribution is parametrized over *the Lie algebra of $G$*. For insights on Lie Groups and Lie algebras, we refer the interested reader to Hall [19]. Thus, $\theta$ specifies the bounds of a uniform distribution in the Lie algebra. It is 6-dimensional, each $\theta_i$ specifies the bounds of the distribution over the subgroup $G_i$: $U(-\theta_i/2, \theta_i/2)$. When the value sampled in the Lie algebra is 0, this corresponds to the transformation $g$ being the identity. The smaller $\theta_i$, the smaller the range transformations in $G_i$ are used, and a dirac distribution on 0 always returns the identity transformation, i.e. no use of the transformation $G_i$.

The value of $\theta$ is learned along the parameters of the network $w$, specifying which transformation is relevant for the task at hand. In the case of classification, the cross-entropy loss is linear and thus expectation can be taken out of the loss.

$$l(\hat{f}_w(x)) = l(\mathbb{E}_{g \sim \mu_\theta}[f_w(gx)]) = \mathbb{E}_{g \sim \mu_\theta}[l(f_w(gx)]. \tag{4}$$

Furthermore, Augerino employs a negative $L2$-regularization on $\theta$, parametrized by $\lambda$, to encourage wider distributions. The resulting training objective is:

$$\min_{\theta, w} \mathbb{E}_{g \sim \mu_\theta}[l(f_w(gx))] - \lambda ||\theta||_2 \tag{5}$$

where a larger $\lambda$ pushes for wider uniform distribution. Everytime an image is fed to the model, Augerino (1) draws a sample from the uniform distribution on the Lie algebra of $G$ (2) computes

---

[7]Note that we apply `FixedSizeCenterCrop`, which resizes the image to 256 on the shorter dimension, before `RandomSizeCenterCrop`. Thus, compared to what is done at training, there is an additional scaling of $256^2/224^2 \approx 1.3$ for the same value $s$.

[8]While in the original paper Benton et al. [6] mentions scale in $x$ and scale in $y$ and shearing, the generators of Aff(2), employed in the paper, in fact correspond to uniform scaling, stretching and shearing.

the transformation matrix in image space through the exponential map (3) augments the image with the corresponding transformation (4) feeds the augmented image to the neural network to perform classification. See Benton et al. [6] for more details on the model.

**Regularizing by transformations**   We noticed that the original code library of Benton et al. [6] disables regularization if any of the 6 coordinates in $\theta$ (one per each of the transformation in Aff(2)) has reached a certain value. From our initial experiments, we understand this is to prevent underflow errors. However, we find it to be too strict if we want to learn multiple transformations at the same time, and thus we shutdown regularization on each $\theta_i, i = 1, \ldots, 6$ when a specific value for that coordinate $i$ is reached.[9] This also allows us to control separately each transformation regularization. While Benton et al. [6] has found that their model was insensitive to the regularization strength, we find it to be a key parameter when applied to ImageNet. This is a first difficulty we face when trying to tackle a real dataset with an equivariant model. While the original model considers always all 6 possible transformations, we also modified the original model to consider any transformation separately and any of their combinations.

### G.2   Augerino on translation discovery

We study if Augerino discovers translation as a relevant transformation to improve performance on ImageNet. We shutdown regularization if the bound of the distributions (separately for each x and y-axis) has reached a value corresponding to $-50\%, 50\%$ translation in image space. We cross-validate between multiple seeds and bounds regularization parameters. As Augerino is employed at validation, we perform testing with 5 different seeds. We also show in Table A4 the results of Augerino disabled at evaluation. Table A3 shows that for different values of the regularization parameter $\lambda$ (see Equation

| $\lambda$ | Top-1 $\pm$ SEM |
|------|-------------------|
| 0.01 | $63.53 \pm 0.0$ |
| 0.1  | $63.78 \pm 0.1$ |
| 0.2  | $64.3 \pm 0.1$  |
| 0.4  | $\mathbf{67.22 \pm 0.0}$ |
| 0.6  | $67.05 \pm 0.0$ |
| 0.8  | $67.01 \pm 0.0$ |
| 1.0  | $66.98 \pm 0.0$ |

Table A3: ImageNet validation set Top-1 accuracy $\pm$ standard error of the mean (SEM) over training seeds, for different $\lambda$.

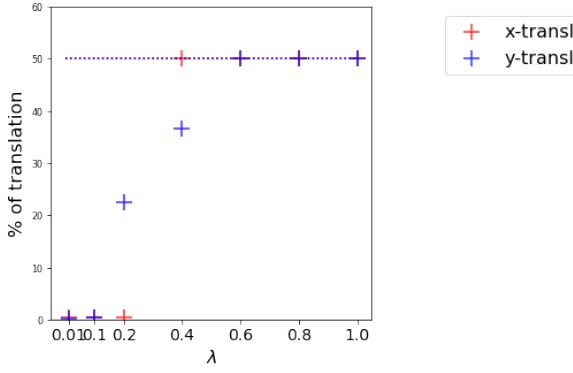

Figure A8: Learned translations versus $\lambda$ values. Every marker correspond to a learned value. Dashed lines correspond to $50\%$ i.e. when the regularization is disabled.

5), different bounds are learned by the model. In Fig. A8 we compare the learned bounds of the distribution. The bounds saturate at a value corresponding in image space to $50\%$, i.e. when the regularization is disabled (shown in dashed lines). This shows that, contrarily to the experiments in Benton et al. [6], the regularization term in Equation 5 strongly impacts the learned bounds. Best performance is achieved for $\lambda = 0.4$, with learned bounds that correspond to sampling a translation in $\approx [-50\%, 50\%]$ on the x-axis, and $\approx [-36.7\%, 36.7\%]$ on the y-axis. Augerino discovers translation as a relevant augmentation, and learn values that improve over the `FixedSizeCenterCrop` (no augmentation) method.[10]

Table A4 shows the performance of the model trained with Augerino for translation discovery, when Augerino is not employed during evaluation time. We do note they are slightly higher (by up to $\approx 1\%$ for $\lambda \geq 0.6$) to the ones reported in Table A8.

---

[9]If the value goes below in the next iteration, regularization is enabled again.

[10]We use Augerino on top of `FixedSizeCenterCrop` pre-processing. For comparison, a model trained with T.(30%) with `FixedSizedCenterCrop` pre-processing achieves $67.08 \pm 0.1$ Top-1 accuracy.

| $\lambda$ | Acc@1 $\pm$ SEM |
|------|-----------------|
| 0.01 | $63.31 \pm 0.1$ |
| 0.1  | $63.49 \pm 0.1$ |
| 0.2  | $64.08 \pm 0.1$ |
| 0.4  | $67.68 \pm 0.1$ |
| 0.6  | $\mathbf{67.98 \pm 0.0}$ |
| 0.8  | $\mathbf{67.95 \pm 0.1}$ |
| 1.0  | $\mathbf{67.99 \pm 0.0}$ |

Table A4: ImageNet validation set Top-1 accuracy $\pm$ standard error of the mean (SEM) for Augerino using translation in x and y axes, for different $\lambda$. These results are when Augerino is disabled at evaluation time (no augmentation of the validation images).

### G.3 Augerino on translation and scale

When we use Augerino on the scale-translation group, we want to control the parameters so that we can use specific shutdown values corresponding to $50\%$ translation and $2000\%$ scaling, and apply translation before scaling as is done in the `T.(30%) + RandomSizeCenterCrop` method. Thus when we use the Augerino model to learn parameters for the scale-translation group, we performed a few modifications to the original code. First, we explicitly compute translation before scaling. Second, Augerino's code uses the *affine_grid* and *grid_sample* methods in Pytorch, where the former performs an inverse warping. That is, for a given scale $s$, the inverse scaling is performed. This does not impact the translation which is performed in both direction (negative and positive translation). For scale, we shutdown regularization if the value $\theta_s$ corresponds to $s = 1/2000 = 0.05\%$, as the inverse scaling will be performed. Third, we want to sample scales corresponding to an increase in size (zooming-in) in order to mimic the effect of `RandomResizedCrop` which selects only a subset of the image. Hence we take $U(-\theta_s/2, 0)$. A sampled value which is negative corresponds to a positive scaling by Augerino given the inverse wrapping. We shutdown the regularization parametrized by $\lambda$ when translation has reached $50\%$ of the width / height, and a scaling of $2000\%$ (the object appears 20 times larger).

Table A5 shows the performance of Augerino when trained on ImageNet with the possibility to learn about translation and scale in conjunction. Best performance is achieved with $\lambda = 0.2$, and we see in Fig. A9 that the model has learned to augment with x-translations only (the red marker being close to $\approx 40\%$ and the green and blue marker being close to 0). Interestingly, when Augerino is disabled at evaluation time, Table A4 shows that the Top-1 accuracy is much higher for large values of regularization compared to Table A5, with the best performing $\lambda = 0.6$. This means that with a more aggressive augmentation, the use of Augerino hurts during inference and only slightly helps during training compared to the no augmentation case (see Table 1a for `FixedSizeCenterCrop`). Still, as in the experiment for translation only, we note that the value of $\lambda$ greatly impacts the results, and that the gain in performance compared to no augmentation is quite small. More importantly, the performance is smaller than when using translation only (see Table A3), which the model could have fallback on if scale was not a relevant transformation. For comparison, a model trained with `T.(30%) + RandomSizeCenterCrop` with `FixedSizedCenterCrop` pre-processing achieves $67.33 \pm 0.0$ Top-1 accuracy. We conclude that while Augerino is a promising model for automatic discovery of relevant symmetries in data, it remains a challenge to apply such methods on a real, large-scale dataset such as ImageNet.

| $\lambda$ | Acc@1 $\pm$ SEM |
|---|---|
| 0.01 | $63.51 \pm 0.1$ |
| 0.1 | $63.73 \pm 0.1$ |
| 0.2 | $\mathbf{63.90 \pm 0.1}$ |
| 0.4 | $63.34 \pm 0.0$ |
| 0.6 | $62.83 \pm 0.0$ |
| 0.8 | $61.0 \pm 0.1$ |
| 1.0 | $57.71 \pm 0.0$ |

Table A5: ImageNet validation set Top-1 accuracy $\pm$ standard error of the mean (SEM) for different $\lambda$.

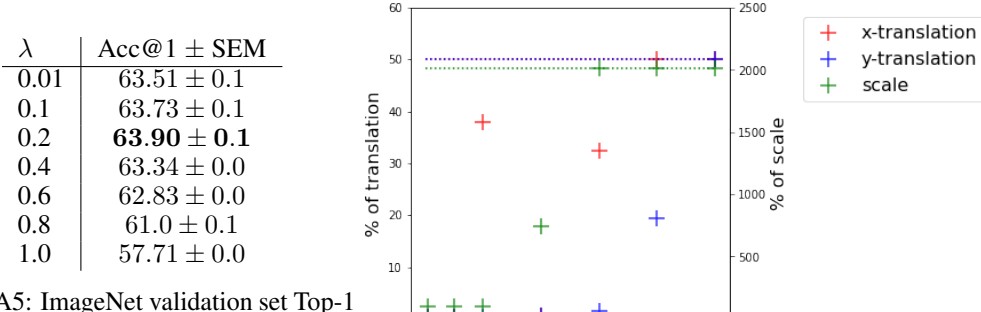

Figure A9: Learned translations versus $\lambda$ values.

| $\lambda$ | Acc@1 $\pm$ SEM |
|---|---|
| 0.01 | $63.11 \pm 0.1$ |
| 0.1 | $63.29 \pm 0.1$ |
| 0.2 | $63.7 \pm 0.1$ |
| 0.4 | $63.40 \pm 0.1$ |
| 0.6 | $\mathbf{64.16 \pm 0.1}$ |
| 0.8 | $63.41 \pm 0.1$ |
| 1.0 | $62.27 \pm 0.1$ |

Table A6: ImageNet validation set Top-1 accuracy $\pm$ standard error of the mean (SEM) for different $\lambda$, disabling Augerino at evaluation.

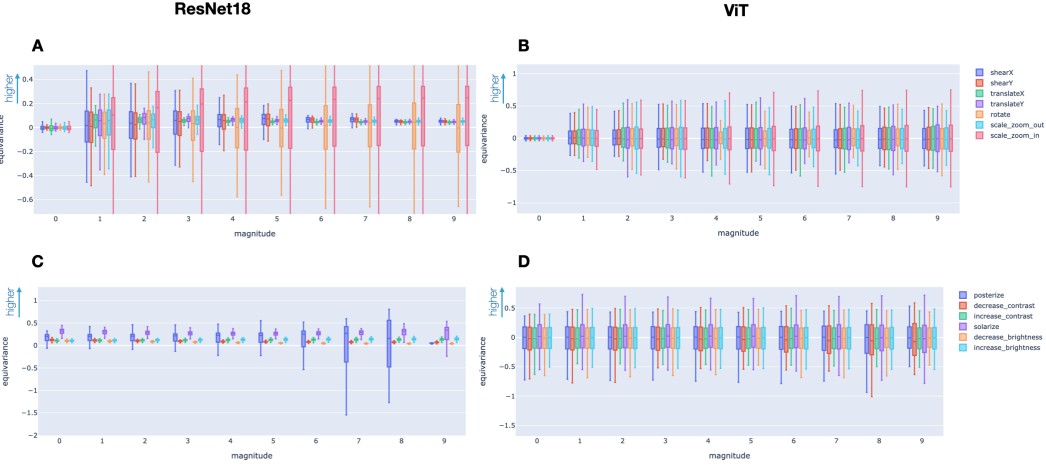

Figure A10: Equivariance for untrained ResNet18 (panel A, C) and ViT (panel B, D). We compare both geometric and appearance transformations using validation set images. We find no significant difference between training and validation set results.