# OpenReview forum: "Grounding inductive biases in natural images: invariance stems from variations in data"
_NeurIPS.cc/2021/Conference — NeurIPS 2021 Poster_

### Official Review · Reviewer_Ybab · 2021-07-13

**Rating:** 7
**Confidence:** 3

**Summary:**

The authors characterize invariances of standard models on ImageNet through the lens of commonly used data augmentation strategies and their relative effects.

**Limitations And Societal Impact:**

Yes

**Main Review:**

The authors provide a thorough and convincing study disproving the existence of many properties that I, and probably many other readers, would have taken for granted. Many very important and surprising things are learned from this paper:
	-- The depth of analysis of the effect of RCC, and of derived augmentations related to it, was very insightful and a rare glimpse into something so ubiquitously used but not often studied.
	-- Viewing the invariances per-class helpfully identify the heterogeneity of the problem and will hopefully encourage per-class solutions rather than the traditional augmentations which are one-size-fits-all vis-a-vis classes.
	-- The discussion of appearance versus geometric invariance is crucial, and this work will hopefully inspire future focus on geometric augmentations that better reflect the underlying non-Euclidean-pixel-space source of these images.
	-- I would have liked to see more about the evaluation of training on one augmentation and then it's impact on other transformations, as for example the appendix figure suggests these effects could be substantial.
	-- It is disappointing that no organizing thoughts could be garnered from the classes where RRC significantly hurt accuracy, as ideally the types of analyses used in this paper could be used to generate better augmentations, but merely identifying them here can be used as a step in that process.

While this study was limited to one kind of model/task on one dataset, and while I think it would be interesting to explore these problems in other contexts that use the same augmentations (e.g. generative models), I view that is definitely beyond the scope of one paper, and thus see no problem with its current scope.

Overall, I found many insightful points that were drawn from highly relevant contexts and think this would be a valuable contribution.


**Time Spent Reviewing:**

1

---

> ### Author Response · Authors · 2021-08-09
> **Response to Reviewer Ybab**
>
> We thank Reviewer Ybab for taking the time to review our paper. We are happy that he/she found our findings thorough, convincing, surprising and of importance. We have indeed tried to provide the most in-depth analysis, e.g. by backing our results with controls experiments and baseline values. We address below their suggestions.
>
> *“training on one augmentation and then it's impact on other transformations”*
>
> We agree with the reviewer that this would be a very interesting follow-up experiment, since we suggest standard augmentations lowers magnitude changes in embedding space.
>
> *“[N]o organizing thoughts could be garnered from the classes where RRC significantly hurt accuracy...”*
>
> We indeed tried to understand if any pattern emerged in the classes hurt by RRC. Looking at the labels classes gave no insight. As we mention in the paper, we compared the similarities (using the most specific common ancestor in the Wordnet tree) among the classes that are consistently hurt with the similarities of these classes and the ones that are either consistently helped or neither helped nor hurt, but did not find any pattern. We agree this is a very interesting question and plan to investigate the factors which impact whether RRC helps or harms performance per-class.

---

### Official Review · Reviewer_b4MW · 2021-07-15

**Rating:** 4
**Confidence:** 3

**Summary:**

The paper explores the links between popular data augmentation techniques and the natural factors of variation present in data, such as translation, rotation, scale, and color. The paper is structured into three main sections, which each describe a set of experiments designed to answer a high level research question. Section 2 is focused on understanding how the popular data augmentation technique Random Resized Crop (RRC) can be decomposed into simpler factors of variation such as translation and scale.  Section 3 proposes a measurement of invariance and studies how the choice of architecture and data augmentation strategy impact invariance to specific transformations.  Section 4 proposes a technique to identify which factors of variation are most prevalent in the data, and finds that, in ImageNet, 1) the factors of variation are class specific, and 2) appearance transformations (like contrast, posterization, brightness changes) account for more variation in ImageNet compared to geometric transformations (translate, scale, shear).

The experiments presented suggest that tailoring augmentation to each class of using more appearance-based transformations might help improve robustness and accuracy.


**Limitations And Societal Impact:**

Yes, the authors discussed both in Section 6.

**Main Review:**

**Major Issues**

1. *Lack of testing for the prescriptions*.  The authors' experiments reveal some interesting directions for improving data augmentation strategies (such as tailoring augmentations per class or using more appearance based augmentations), but the authors don’t test these strategies themselves.  The experiments and experimental techniques used by the authors would be more valuable if the authors could prove that they lead to beneficial prescriptions.

2. *Unclear how decomposition of RRC is helpful*.  The authors show that RRC can be decomposed into a scale and translation, but find that neither augmentation alone can replace RRC.  So, similar to the previous issue, it is not clear why knowing which factors of variation a data augmentation technique corresponds to if we cannot use this knowledge to improve performance.

3. *Role of the scale transformation*. In Section 2, it's surprising that the contribution of scale was relatively minor in the performance of RRC. Possibly, the center crop might miss the object of interest so we end up scaling a piece of the image that doesn't correspond to the class at all.  In contrast, for the translation, the fixed size random crop selects a random location of the image, so it has some chance of randomly selecting the part of the image that is relevant for the class.  The authors could look at whether scale helps certain classes where the object is less likely to be cut out of the image (such as the textile or scenery classes). Or, using a dataset with bounding boxes available like the ILSVRC2013 detection dataset (https://image-net.org/challenges/LSVRC/2013/index.php), the authors could ensure that some fraction of the bounding box overlaps with the crop before scaling.

4. *Synthetic factors of variation*. The set of transformations that the authors explore are limited in that they can all be written as a coded function that can be applied to the images. The robustness community has started to refer to this type of transformation as “synthetic”. It has been shown that synthetic augmentations often do not help on more realistic or natural distribution shifts [1]. It would be interesting if the authors tried to explore some way to capture more natural factors of variation, which could be discovered through human labeling.

**Minor Issues**

5. Not clear where the results are for the experiments described in lines 195-201
6. Distinction between appearance and geometric transformations should be more clearly described or illustrated in main text
7. Difficult to read figures – text is quite small.

**Originality**: The work has novel experimental insights.

**Quality**: See comments above.

**Clarity**: The writing is sometimes hard to follow, but the main points are still understandable.  There are too many rhetorical questions sprinkled throughout.

**Significance**: The experimental findings would be more significant if the authors could show they resulted in prescriptions that could improve model performance and accuracy.




**Time Spent Reviewing:**

3 hours

---

> ### Author Response · Authors · 2021-08-09
> **Response to Reviewer b4MW**
>
> We thank Reviewer b4MW for taking the time review the paper. Below, we address the main comments.
>
> *Lack of testing for the prescriptions.*
>
> Though we agree that practical recommendations are certainly valuable, the goal of our work, as other reviewers note, is to study “many properties that [reviewer Ybab], and probably many other readers, would have taken for granted”—not to provide prescriptive, performance-oriented statements. We substantiate our conclusions with carefully designed controls (e.g., across versus per-class), error bars for estimates (e.g., we plot distributions when possible and report standard-errors), and configurations (e.g., single transformation vs. sub-policies). We tried to provide the most thorough analysis of our findings (e.g. using WordNet tree hierarchy to study if any pattern emerged between classes hurt by RRC).
>
> However, as an initial test of how these observations might motivate prescriptive changes, we ran initial experiments using the top class-specific augmentations derived from similarity search. While these experiments did not improve over the classic augmentation approach, we emphasize these were initial and we believe further research could unlock the potential of per-class augmentations. Consistent with this, Hauberg et al., [19 in paper] demonstrated that, for a synthetic colored MNIST experiment, using different augmentations per class improved performance. While discovering corresponding augmentations for naturalistic images will likely be challenging, we believe that approaches where augmentations are learned hold promise for this direction.
>
> *“Unclear how decomposition of RRC is helpful. ”*
>
> Our goal with this analysis was to understand the relative importance of translation and scale to the performance gains induced by RRC. Notably, we observed that despite the supposed built-in translation invariance of CNNs, translation augmentation was substantially more important than scale, suggesting that learning invariance to translation is critical for high-performance models. As reviewer Ybab notes, this is a “very insightful and a rare glimpse into something so ubiquitously used but not often studied” 2) illustrates that several classes are consistently hurt by this commonly used augmentation. Both observations that run counter to common practices in computer vision.
>
> *“Role of the scale transformation.”*
>
> We thank the reviewer for the suggested experiment to check whether the center crop may miss the object of interest relative to a random crop of the same size. However, we think there is substantial evidence that this is not the case. First, imagine that across the dataset, the center of mass of the object of interest is uniformly distributed across the entire image. In such a case, a center crop of a given size would have the same probability of containing the object as any other crop of the same size. Regardless of the object’s position, the probability that the crop contains the object of interest would be (area of crop)/(total image area), such that the position of the crop should be irrelevant in expectation.
>
> In contrast to the hypothetical above, however, we know that ImageNet objects are not uniformly distributed across the image, but rather are strongly biased towards the center of the image. We investigated this in the appendix of our paper, showing in Figure A6 that the center of the object of interest in ImageNet (defined as the image foreground) was very close to 0.5 for both x and y coordinates. The center crop should therefore contain the object of interest for the vast majority of images.
>
> *Synthetic factors of variation*
>
> We agree with the reviewer that natural factors of variation are hard to interpret, measure, and compare. Consequently, prior work has primarily focused on analyzing variations only in purely synthetic settings in which the data as well as the transformations are synthetic. To try to generalize these ideas to more realistic settings, we focused on a real dataset (ImageNet) with a simple, interpretable, and measurable set of transformations. While we agree that focusing on more realistic transformations would be extremely interesting, this is a substantially more challenging task as it’s not clear how to best define on-manifold image transformations. We therefore argue that understanding real images in the context of simpler transformations is a necessary step to eventually tackle more complex transformations.
>
> We thank Reviewer b4MW for his minor issues suggestion and will include them in the next version of our work.

---

> ### Author Response · Authors · 2021-08-20
> **Has our response addressed your concerns?**
>
> Hello reviewer b4MW,
>
> We’d be grateful if you can confirm whether our response has addressed your concerns. We’d be glad to answer any outstanding questions. To recap, in our response we:
>
> - Acknowledge the value of practical recommendations and highlight the value of non-performance oriented research. We emphasize the substantial value in verifying, expanding on, and elucidating the underlying mechanisms of existing observations.
> - Highlight several practical applications spurred by our work including
> 	1. the consistent performance degradation of RRC for some classes
> 	2. the source of invariance is not primarily due to architecture or data augmentation
> 	3. the class-specific nature of factors of variation, counter to how augmentations are applied universally
> - Thank reviewer b4MW for the suggestions and agree to incorporate these improvements

---

### Official Review · Reviewer_WyYA · 2021-07-18

**Rating:** 5
**Confidence:** 5

**Summary:**

This paper provides an analysis of the invariance of neural networks (ResNet) trained on image classification (ImageNet) to image transformations, and of the role of data augmentation in encouraging invariance. The paper provides different types of results. First, in Section 2, the authors analyse what components of the data augmentation scheme _random resize crop_ (RRC) mainly drive the gains in performance, and conclude that translation is the main transformation. Second, in Section 3, the paper discusses the invariance distribution (measured as the relative cosine distance between augmented and original images) of models to several transformations. The paper presents results of untrained models, trained with and without augmentation, in order to isolate the sources of invariance. The authors conclude that untrained models already exhibit certain level of invariance, training improved the invariance to some (but not all) transformations, and data augmentation further improved invariance, but only slightly and not always. Finally, in Section 4, the authors aim to characterise the factors of variation within ImageNet comparing the similarity of samples under some pre-defined transformations. This section presents multiple results, the ones highlighted by the authors being that factors of variation differ across classes and that training a model increases the invariance to the transformations that are found to explain most variation in ImageNet.

**Ethical Concerns:**

I see no relevant ethical concerns specifically derived from this work, but it would be worth mentioning, in my opinion, the concerns associated with _any_ work that makes use of ImageNet. See, for instance, [Prabhu and Birhane (2020)](https://arxiv.org/abs/2006.16923).

_Note: I have updated this note after the original review was submitted._

**Limitations And Societal Impact:**

The authors do mention at the end of the paper some of the limitations of their work, as well as state that they do not see potential negative societal impacts.

**Main Review:**

_Edit, post-rebuttal: I have upgraded my numerical evaluation by one point_

---

## Strengths or contributions

I highlight that the paper addresses a topic that has received increasing attention in the last couple of years, namely data augmentation and invariance to image transformations, and therefore it has the potential of being of interest to the community. The paper offers a wide set of results and some of them are interesting and insightful. For example, the most relevant results to me are those presented in Section 3, which analyse the invariance of models separately to various transformations.

## Weaknesses and justification of overall score

I begin by summarising the justification of my overall evaluation in terms of the four main dimensions of the review: in terms of *quality*, I think the paper is technically correct, the methods are justified and sufficient details are generally provided (with some exceptions that I mention below). The paper is also *clearly written* for the most part and is easy to follow. As an exception, I note that Section 4 is a little confusing, perhaps because many results are tried to be presented in a limited space. As a weaker aspect, while the paper provides some interesting results, I would argue that the overall *significance* is limited. In my humble opinion, there are no major findings that offer new perspectives or insights about data augmentation and invariance, but rather a collection of results that either confirm findings from previous work or offer limited new insights. In terms of *originality*, the methods used are mostly standard or slightly different from other variants in the literature. In the remainder of my review I elaborate on these weaker aspects.

The main part of the paper is divided into three sections (2, 3 and 4) that present significantly diverse analyses and results. As a general comment, I believe that such breadth of points of analyses may dilute the main messages and compromise the depth of the most interesting ones. For example, in my opinion, the most relevant results are those in Section 3, but the conclusions could be much stronger with an analyses of additional architectures and data sets. I will comment on each Section separately.

### Section 2, analysis of `RandomResizedCrop`

I think the significance and relevance of the results in this section are quite limited. The section refers, in the title and in the text, to an analysis of the composition of standard data augmentation *techniques* (in plural), but in fact it is an analysis of random resized crop (`RandomResizedCrop`, RRC). While an in-depth analysis of multiple transformations could be of interest, the results of one transformation on one architecture and one data set are, in my opinion, of little relevance.

Furthermore, the conclusion, namely that translation augmentations plays a major role in improving the performance, should provide little insights to those familiar with the computer vision literature. While RRC may have gained popularity in recent years, a very common and simple augmentation scheme and widely known to provide very good results has been the combination of horizontal and vertical translations and horizontal flips. See for example [1, 2]. The results of using a $\beta$ distribution instead of uniform sampling for the augmentation parameters are also expected and, in fact, previous work has reported results of training with data augmentation with different sampling than uniform with slight improvements.

### Section 3, invariance of architectures and augmentations

As mentioned before, I find this section the most interesting and relevant. In particular, I am not aware of other articles that have analysed the role of individual transformations on model invariance. However, the main conclusions highlighted by the authors are more or less known from the literature. For example, that convolutional neural networks do not exhibit sufficient invariance to translations, despite the widely spread claim, has been noted by multiple authors before, perhaps most clearly by [3] (as this paper cites in the related work section). Nonetheless, in Section 3 this observation is qualified as "surprising".

The other main observation of the section is that data augmentation barely increases the invariance of the models, also highlighted as surprising. However, this has also been reported before, for example by [4], with a related but different invariance score.

Finally, I have technical question regarding the invariance metric: it is not entirely clear what the baseline $b$ is in Equation (1). What are $x_i$ and $x_j$ in its definition (line 170)? Two random samples? This is a crucial aspect that I have not fully understood from the paper, although I have assumed the high-level intuition is clear.

### Section 4, factors of variation in ImageNet

This section contains multiple results, some of them of interest, but the amount of different experiments and results seems to compromise the readability of the section. For example, some conclusions are to be interpreted from figures in the main paper, some from figures in the appendix and some from results that are not displayed in figures but only reported textually in a paragraph. As I mentioned above, the main messages may be diluted among the variety of minor results and conclusions.

On a more technical level, it is not surprising that transformations do not measurably increase the average similarity between pairs, since the factors of variations in natural images are much more complex. In other words, the goal of trying to disentangle the factors of variation in ImageNet is of interest, but it is perhaps too ambitious to be tackled as a section of a paper, by measuring the similarity explained by a limited set of transformations.

Finally, like in other sections discussed above, the main conclusions do not provide substantial insights. For example, that training improves the invariance to the transformations that drive the most similarity is to be expected. In contrast, the conclusion about class-specificity of some transformations is interesting and could have practical impact, but the analysis provided in this paper is not deep enough and is left as future work.

## Other questions

Here are some other comments, questions or suggestions that are more minor or disconnected from the main arguments of my review. These had a nearly negligible impact on my overall assessment and are included here for completeness:

* In the first paragraph of Section 3.2, it is not clear that the conclusions refer to the "Untrained" box plots of Figure 2. Furthermore, it is particularly confusing that the text reads "models **learned** to be invariant", while we are talking about _untrained_ models.
* The notion of invariance in neural networks changes across papers. For example, some consider invariance as measured by the classification output of a model (which should probably refer to as robustness), some measure invariance at the softmax output, and more generally, some measure invariance at a high-level embedding. It would be positive to define the notion of invariance that the authors use early enough in the introduction.
* In Section 4, why is AutoAugment used, instead of the set of transformations analysed in Section 3?
* The last paragraph of Section 3.2 is confusing. Is it a summary of what has been said before? If so, why do the authors talk here about _equivariance_ instead of _invariance_? Is it a discussion of new results beyond Figure 2? If so, where are these results?
* Do the authors have an explanation for the higher invariance to rotations of untrained models?

## Typos

Below I list some potential typos or language errors:

* Line 169: [13] [31] -> [13, 31]
* Line 211: access to generative model -> access to a generative model
* Line 218: effect data augmentation -> effect of data augmentation

## References

* [1] Goodfellow et al. Maxout networks. ICML. 2013.
* [2] Springenberg et al. Striving for simplicity: The all convolutional net. ICLR. 2014.
* [3] Zhang. Making convolutional networks shift-invariant again. 2019. ICML
* [4] Hernandez-Garcia et al. Learning robust visual representationsusing data augmentation invariance. 2019. https://openreview.net/forum?id=B1elqkrKPH

**Time Spent Reviewing:**

6

---

> ### Author Response · Authors · 2021-08-09
> **Response to Reviewer WyYA**
>
> We thank Reviewer WyYA for taking the time to review our paper. We agree that identifying the source and role of invariance is important for the research community.
>
> While Reviewer WyYa’s main concern regards the significance of our results, arguing that our work provides limited new insights or perspectives, we would like to emphasize that two other reviewers noted that our findings on the topics are "surprising"[Ybab], "packed with meaningful results" [dbox], and "disprov[e] the existence fo many properties that I, and probably many other readers, would have taken for granted" [Ybab].
>
> ### Section 2
>
> We thank the reviewer for the references illustrating the utility of translation and will include those in our discussion.
>
> *Focus on RandomResizedCrops (RRC)*
>
> We note that we chose to focus on RRC because they are the de facto data augmentation approach (along with horizontal flips) used in the training of ImageNet scale models. However, we agree with the reviewer that the title of Section 2 is framed too broadly given our focus on RRC. We will update the title of Section 2 to narrow the scope to focus on RRC.
>
> Our analysis of RRC 1) decomposes the interactions of translation and scale. We show not only the gain from translation augmentation but also the unexpected small improvement when using scaling in addition. As Reviewer Ybab notes, this is a “very insightful and a rare glimpse into something so ubiquitously used but not often studied” 2) illustrates that several classes are consistently hurt by this commonly used augmentation. Both observations run counter to common practices in computer vision.
>
> *“previous work has reported results of training with data augmentation with different sampling than uniform with slight improvements.”*
>
> Previous work does show a slight improvement over a uniform distribution. In contrast to prior work, however, our goal was not to use the $\beta$ distribution to improve performance over uniform, but rather to understand the tradeoff between injecting variability and mitigating the test-train discrepancy [5]. By measuring this tradeoff, we illustrate the sensitivity of RRC to the choice of distribution as well as the precise balance achieved by the common default parameters. Indeed, as we explain in the paper, very small values of $\beta$ lower the train-test discrepancy but decrease performance, as they do not encourage scale invariance by sampling scales near 1, thus barely augmenting on scale (low variance of the distribution over 1/s).
>
> ### Section 3
>
> *“However, the main conclusions highlighted by the authors are more or less known from the literature.”*
>
> We thank the reviewer for the reference to [4] which our results confirm through a different methodology. We agree our work is not the first to imply that CNNs lack the expected translation invariance and cite [3] in our related work. However, we would argue that there is substantial value in verifying and expanding on this observation with different approaches given the counter-intuiveness of this observation. Consistent with this perspective, reviewer dbox points out “recent works indicating that the inductive bias of CNNs may not be as essential or well-understood as previously thought, and this may be one of the clearest analyses of this observation to date.”
>
> Furthermore, our analysis traces the source of invariance and further studies inherent *equivariance* properties across architectures (ViT and ResNet-50). We illustrate that CNNs do have inherent translation *equivariance* due to their architecture in contrast to transformer models (ViT). Furthermore, we show invariance stems from the data itself rather than the choice of architecture or data augmentation. To our knowledge, both of these observations are novel in the literature.
>
> *“[I]t is not entirely clear what the baseline b is in Equation (1). What are xi and xj in its definition (line 170)?”*
>
> To ensure our invariance measure does not capture noise in embeddings or the effect of the specific transformation applied, we define our baseline b to capture the distances we would observe by chance in our embedding space. Therefore, our baseline, b, measures the distance of randomly selected samples with randomly sampled transformations applied. In Equation (1), we illustrate the definition of the baseline for two randomly selected samples xi and xj. The closer our invariance measure is to 1 the more invariance to the transformation relative the baseline. We agree the notation in our definition should be more clear and we will change it accordingly.
>
> ### Section 4
>
> *“On a more technical level, it is not surprising that transformations do not measurably increase the average similarity between pairs, since the factors of variations in natural images are much more complex.”*
>
> We agree that the transformations present in natural images are extremely complex and that it is not a priori obvious that simple transformations would capture these variations. However, while it is perhaps unsurprising that simple transformations cannot increase similarity across the entire dataset, what is surprising is that the same transformations consistently increase average similarity among pairs when *examined per class*. This suggests factors of variation are class-specific, despite data augmentation applying transformations universally across classes.
>
> *Finally, like in other sections discussed above, the main conclusions do not provide substantial insights. For example, that training improves the invariance to the transformations that drive the most similarity is to be expected.*
>
> We’d like to emphasize several substantial and, to our knowledge, novel insights from Section 4:
>    1) Appearance variation is more prevalent compared to geometric, despite the focus of augmentations on geometric transformations (Reviewer Ybab found this result “crucial” and suggested “will hopefully inspire [the] focus [of] future [work]” ).
>    2) RRC, a commonly used data augmentation, consistently hurts performance for a subset of classes by as much as 22% top-1 accuracy (effect is consistent across random seeds).
>    3) Variation, as noted above, is class-specific—running counter to how augmentations are applied universally across classes. We further show in Section 4.5 that similar classes, measured via the WordNet hierarchy, share similar factors of variation.
>
> ### Other questions
>
> *“In the first paragraph of Section 3.2, it is not clear that the conclusions refer to the "Untrained" box plots of Figure 2. Furthermore, it is particularly confusing that the text reads "models *learned* to be invariant", while we are talking about untrained models."*
>
> We agree with the Reviewer that the first paragraph of Sec. 3.2 is confusing, as we discuss both untrained and trained models. The reference to “models learned to be invariant” refers to trained models. We will improve the wording and clearly separate discussion of trained vs untrained models.
>
> *Definition of invariance*
>
> We thank the Reviewer for the suggestion to describe which layer we use to examine invariance/equivariance earlier in the paper and will do so.
>
> *“In Section 4, why is AutoAugment used, instead of the set of transformations analysed in Section 3?"*
>
> We use AutoAugment in Section 4 to include a broad set of transformations without the combinatorial explosion of all transformation types, parameter choices, and image pair samples.
>
> *“The last paragraph of Section 3.2 is confusing.”*
>
> We agree the wording is potentially confusing and will clarify we discuss equivariance (a more general property than invariance) in this section. We include a discussion of equivariance to capture possible predictable responses to transformations beyond invariance. We summarize our methodology and results for equivariance in Section D and Figure 14 of the Appendix.
>
> *“Do the authors have an explanation for the higher invariance to rotations of untrained models?”*
>
> We conjecture training decreases rotation invariance based on the object-centric bias of the ImageNet dataset. We hypothesize that if most objects appear in a canonical pose, training would incentivize models to use rotation as a signal (for example, bats are more likely to appear upside down than automobiles) which would decrease invariance to rotation over training. However, we have not verified this assumption which is why we did not include this conjecture in the paper. We agree, however, that this would be an interesting question to explore further.
>
> [5] Touvron, Hugo, Andrea Vedaldi, Matthijs Douze, and Hervé Jégou. “Fixing the Train-Test Resolution Discrepancy: FixEfficientNet.” ArXiv:2003.08237 [Cs], November 18, 2020. http://arxiv.org/abs/2003.08237.

---

> > ### Comment · Reviewer_WyYA · 2021-08-26
> > **Rebuttal discussion #1**
> >
> > Dear authors,
> >
> > I apologise for the delay in my answer to your rebuttal. I hope you understand that reviewers are heavily loaded this year and the timing is not ideal.
> >
> > Thank you for your responses, and for highlighting some comments of in the reviews of my fellow reviewers, which I have carefully read.
> >
> > I was glad to read that the results in the paper were "surprising" to Reviewer _Ybab_. As reflected in my review, most results were not too surprising to me, perhaps because my own research is very close to these topics. In any case, results can be surprising and have little or no significance, as well as the other way around.
> >
> > I actually mostly agree with Reviewer _dbox_ in that the paper is "packed with meaningful results". However, as I mentioned in my original review, I see the fact that the paper is packed with results as weakness. In my humble assessment, I think the paper offers a collection of many results (some of which have been discussed in previous work), but it falls short at weaving a cohesive web with a distinct, overarching message. To illustrate, these are the findings highlighted by Reviewer _dbox_:
> >
> > > * ViT and ResNets have similar invariance properties, and ViT may even exhibit better invariance (i)
> > > * Data augmentation is specific to each class and can even hurt performance for some when applied globally (ii)
> > > * Translation and cropping augmentations interact to produce better performance when applied together (iii)
> > > * Geometric transformations may be the least relevant to the primary factors of variation in ImageNet (iv)
> >
> > What is the overarching message? (i) and (ii) are interesting (to me ii in particular, as I have mentioned), but the paper falls short as shedding sufficient light and instead studies other less related aspects. That translation and cropping work better together (iii) is something we have known empirically for years. Like I mentioned, translation and scaling (mostly equivalent) have been common augmentation techniques in the literature. (iv) is interesting, but also not studied in depth.
> >
> > In the remaining of this answer, I follow up the specific comments.
> >
> >
> > ### RandomResizedCrops (RRC)
> >
> > I appreciate that you consider changing the title of the section, which will probably more faithfully reflect the content. However, my concern regarding the significance is mostly about the content being limited to `RandomResizedCrops`. I do not deny the interest of the experiments, but simply have the opinion that the significance is limited as is currently analysed.
> >
> > ### Invariance of architectures and augmentations
> >
> > I agree with you that there is value in "verifying and expanding" insights from previous work. Do you agree that such previous work should be discussed and the claims of novelty adapted accordingly? In this regard, please note that in my assessment of the paper I am not denying the value of the contributions or qualifying them as useless, bur rather putting them in context and forming an overall assessment of the significance based on them. I hope my first review, where I discuss both strengths and weaknesses, reflects this faithfully.
> >
> > Thank you for the clarification of Equation 1. Having understood that the baseline aims to characterise a "random" distance between samples in the embedding space, then I note that your metric of invariance is closely related to the invariance score in [[4](https://openreview.net/forum?id=B1elqkrKPH)] (as numbered in my review), Equation 2, where the formula is the same but the baseline is the average pairwise distance. Do you agree?
> >
> > ### Factors of variation in ImageNet
> >
> > As I have discussed, the class-specificity of data augmentation is an interesting finding, but the paper only superficially analyses this and leaves is for future work. Again, I acknowledge this contribution, as well as the other results mentioned, but I see them as a collection of multiple rather superficial results.
> >
> > ### Other questions
> >
> > I appreciate the answers to some questions and your willingness to incorporate some of the feedback in the paper.

---

> > > ### Author Response · Authors · 2021-08-31
> > > **Response to Rebuttal Discussion #1**
> > >
> > > We thank Reviewer WyYA for the reply to our response.
> > >
> > > *“Lack of a cohesive story“*
> > >
> > > We respectfully disagree that our results are not combined into a single cohesive goal. Our work aims to answer the following challenge: discovering the factors of variation in real images and linking these to the invariances learned by a diverse set of model architectures (with or without the use of standard data augmentation). Understanding and predicting the responses to changes in factors of variation is critical for out-of-domain generalization [1,2]. To explore this challenge, we first study RRC, a standard data augmentation technique believed to instill invariance. Next, we systematically measure the invariance of various architectures and probe its source. Finally, we uncover the factors of variation in realistic datasets—most of the previous work focused on synthetic datasets—and link those factors to the invariance we observe in large scale vision models. All of these results together help to address the foundational question of whether models encode the variation in natural images.
> > >
> > > We address specific sections below.
> > >
> > >
> > > ### Study of `RandomResizedCrop` (RRC)
> > >
> > > Thus, we start by studying the standard data augmentation technique: RandomResizedCrop. Hence we agree “translation and scaling (mostly equivalent) have been common augmentation techniques in the literature”: this is the reason for our choice, and our work shows that the gain from scaling in addition to translation is very minor, which we have not seen in the existing literature. Given that RRC are used in almost all vision models in some form or another, we respectfully disagree that our study of RRC limits the significance of our work.
> > >
> > > ### Invariances of architecture and augmentations
> > >
> > > Then, we move to the study of the invariances learned by models. We acknowledge existing literature in our paper and cite them when appropriate. We agree our formula for the similarity change is linked to [4] and we thank Reviewer WyYA for pointing out—we’ll add it to the related work, however note that our baseline is different than theirs as we also consider the effect of a random transformation $T_\theta$ (not only random pairs), to account for the change induced by general transformations on the embedding space. Furthermore, we study these invariances in Vision Transformers, finding that they surprisingly feature substantial translation invariance. To our knowledge we are the first to study these questions in transformer based models for vision.
> > >
> > > ### Factors of variation in ImageNet
> > >
> > > Finally, we tackle the challenge of uncovering variation in natural images (via ImageNet)—a task previous work only addresses in the synthetic setting. We find variation is class-specific, mostly related to appearance transformations, and aligns with the invariances learned by models (with or without data augmentation).
> > >
> > > We respectfully disagree with reviewer WyYA that our analysis of the factors of variation is “superficial”. To probe whether variation is class-specific we conducted several controlled analyses decomposing differences among image pairs via a large collection of transformations, both geometric and appearance-based. We illustrate a systematic difference in how well top transformations explain variation across all versus per class both via aggregate averages (with error bars) and in distribution (plotted in Figure 3). We also qualitatively examine the specific transformations illustrating for example a clear pattern among textile classes: textiles have no canonical pose or color, matching our intuition (Section 4.5). We then further analyze this pattern by measuring systematically across all classes the extent to which similar classes share factors of variation (Figure 4). We find similar classes do indeed share factors of variation by repeating our analysis on combinations of two variation rank similarities and three measures of class similarity via the WordNet Hierarchy (Wu-Palmer score, Leacock-Chodorow Similarity, and path similarity). We believe these experiments are comprehensive and constitute ample evidence for the insightful conclusion that variation is class-specific.
> > >
> > >
> > > [1] Yoshua Bengio, Aaron Courville, and Pascal Vincent. Representation learning: A review and new perspectives. IEEE transactions on pattern analysis and machine intelligence, 35(8): 1798–1828, 2013. Publisher: IEEE.
> > > [2] Karel Lenc and Andrea Vedaldi. Understanding image representations by measuring their equivariance and equivalence. International Journal of Computer Vision (IJCV), 127(5), 2019.

---

> > > > ### Comment · Reviewer_WyYA · 2021-09-01
> > > > **Final comments**
> > > >
> > > > Dear authors,
> > > >
> > > > Thank you for your additional comments, which I have carefully read. As I mentioned in my review and the responses, I do acknowledge the contribution of the multiple experiments and results, which you have emphasised in your last comment. I also see the relationship between the different parts. However, when I form an overall assessment of the paper, the main weaknesses I see are 1) in terms of significance and originality, that the paper provides some by limited contributions to our current understanding of the role of data augmentation, invariance and the factors of variation on natural images, leaving the investigation of the most interesting findings as future work, and 2) in terms of clarity, that the paper has room for improvement to form a more cohesive set of experiments where all pieces shed light on the same direction.
> > > >
> > > > Still, these weaknesses mentioned, I also don't think the paper is deeply flawed, and I will therefore not strongly argue against its acceptance.

---

> ### Author Response · Authors · 2021-08-20
> **Has our response addressed your concerns?**
>
> Hello reviewer WyYA,
>
> We’d be grateful if you can confirm whether our response has addressed your concerns. We’d be glad to answer any outstanding questions. To recap, in our response we:
>
> - Outline how our findings provide new results and note other reviewers found our results both meaningful and surprising. We believe our contribution would be surprising and meaningful to the research community
> - In particular highlight our
> 	1) novel investigation of RRC and results showing RRC consistently hurts performance for some classes. In contrast to previous work, we decompose the interactions of translation and scale in RRC, noted by reviewer Ybab as a “very insightful and rare glimpse into something so ubiquitously used but not often studied.” Finally, in contrast to previous studies of distributions over RRC, we quantify a tradeoff between injecting variability and mitigating the test-train discrepancy.
> 	2) measure of the role of inductive biases of CNNs via invariance/equivariance stands out from previous work as “the clearest analysis of this observation to date” as noted by reviewer dbox
> 	3) surprising characterizations of factors of variation in ImageNet such as the prevalence of appearance transformations and discovery that variation is class-specific—counter to how augmentations are universally applied
> - Thank reviewer WyYA for the suggestions and agree to improve the terms defining equi/invariance and clarify the wording in Section 3.2

---

### Official Review · Reviewer_dbox · 2021-07-22

**Rating:** 8
**Confidence:** 4

**Summary:**

In this work, the authors explore to what extent learning, data augmentation, and architectural inductive bias (ResNet18 and ViT) impact invariance to different sets of transformations for ImageNet-trained classifiers. They measure this invariance as cosine distance between an image and its transformed counterpart. They further probe the transformations that explain variation in ImageNet by designing a metric to measure how transformations affect inter-image similarity (pairs of different base images where one is transformed). A higher score on this metric is said to indicate stronger correspondence between the transformation and factors of variation in the data.

The authors find in general that data augmentation and architectural inductive bias have measurable (and decomposable) but weak affects on the invariances exhibited by the models, and also that these invariances are only weakly aligned with the major variation in the data, leading them to argue that the main source of invariance must be the data.

There list of worthwhile component findings in the paper is large. Just a few interesting ones that stuck out to me were:

- ViT and ResNets have similar invariance properties, and ViT may even exhibit better invariance
- Data augmentation is specific to each class and can even hurt performance for some when applied globally
- Translation and cropping augmentations interact to produce better performance when applied together
- Geometric transformations may be the least relevant to the primary factors of variation in ImageNet

**Limitations And Societal Impact:**

The authors should discuss the potential impact of the claim that data is more responsible than the model design in determining invariance, because if true it would imply that researchers may have a stronger responsibility to attend to dataset construction than to modeling when considering potential negative outcomes of a model. This isn't so much a negative impact of this particular work as it is informative about how researchers should assess negative impact of their own work.

**Main Review:**

I found this paper quite interesting and packed with meaningful results. I have some doubts about whether the metrics proposed won't end up misleading in the end, but I also understand that such metrics are difficult to validate. There have been number of recent works indicating that the inductive bias of CNNs may not be as essential or well-understood as previously thought, and this may be one of the clearest analyses of this observation to date. My comments and criticisms are mostly minor:

- I read the following two sentences from the same section as contradictory: (1) "Surprisingly, untrained ViT models also featured stronger invariance to translation when compared to untrained ResNet18 (Fig. 2a, b, compare orange with light blue), suggesting that despite the convolutional inductive bias present in ResNet18, translation invariance is more prominent in ViT models." (2) "We find equivariance to translation for untrained ResNet18 that is absent for ViT, highlighting the architectural inductive bias of CNNs to translation." This should be cleared up as it's an important finding in the paper.

- The authors argue that ResNets/ViTs have similar invariances despite "markedly different inductive biases", but is this true? Don't the findings provide evidence that architectural differences are misleading as biases if their generalization properties turn out to be quite similar?

- If class-level augmentation doesn't help performance in the end, what should we make of the relevant results in the paper? Could they be spurious? Does it help with a minority of difficult validation images?

- typo: "prose" --> "pose"

**Time Spent Reviewing:**

5

---

> ### Author Response · Authors · 2021-08-09
> **Response to Reviewer dbox**
>
> We thank Reviewer dbox for carefully reviewing our paper. We are very happy that the reviewer found the paper quite interesting, and that we provide a large number of worthwhile findings. Below we address the minor comments:
>
> *“I read the following two sentences from the same section as contradictory...”*
>
> We agree the wording of this section is unclear. Please note that we start by discussing invariance (e.g., for translation function $T(x)$, $f(T(x)) = f(x)$), and find that untrained ViT models have more translation invariance than untrained ResNet18. The second part discusses equivariance, a more general property than invariance in which a transformation in the input corresponds to an equivalent transformation of the output (e.g., $f(T(x)) = T(f(x)$). We note that invariance is a specific case of equivariance and that equivariance does not necessarily imply invariance (though invariance does imply equivariance). We will elaborate on this distinction in the main text.
>
> *“The authors argue that ResNets/ViTs have similar invariances despite "markedly different inductive biases", but is this true?"*
>
> We agree with the reviewer that our findings demonstrate that the differences in architectural inductive biases play a minor role in the overall generalization behavior of the trained models and that they can be misleading. However, we agree that the way this is phrased can be confusing. We will clarify that we meant that models arrive at similar invariance properties despite “markedly different *architectural* inductive biases” to distinguish between the learned inductive bias and the built-in bias and to clarify that these architectural differences are not critical for the resultant generalization properties.
>
> *“If class-level augmentation doesn't help performance in the end, what should we make of the relevant results in the paper?”*
>
> While we did make initial attempts at leveraging per-class augmentations to improve performance, we emphasize that there are many ways to implement and determine per-class augmentations and that our initial investigation was fairly limited: we primarily investigated using the transformations discovered by similarity search. We expect that more sophisticated approaches which aim to learn per-class augmentations directly from data will be more successful, but require substantial engineering effort, and we therefore view them as out of the scope of the current work.
>
> *“Could they be spurious?”*
>
> While preliminary experiments applying top transformations from similarity search did not increase classification performance, we are confident in our findings that the impact of augmentation varies across classes. We base our conclusion on several experiments. First, we compare the distribution of similarity changes for the top transformations on the aggregate against those per class. We plot the full distribution of all top transformations per class in Fig. 3B both for the networks trained with and without the standard augmentations, showing per class similarity changes consistently lie above zero for all classes, in contrast to those for the aggregate top transformations. We also measure the mean similarity change across and per class to further illustrate this difference and report standard errors. Second, we show RRC consistently hurts a subset of classes by as much as 22% top-1 accuracy. We include details of our analysis, consisting of 25 pairwise comparisons across 5 runs, in Section 4.3 and Appendix D1. Notably, the classes harmed by RRC were consistent across random seeds, suggesting that this result was not merely a consequence of training noise. Finally, we confirmed the top class-specific transformations correspond to intuitive factors of variation with similar classes exhibiting similar factors of variation (Section 4.4). Nevertheless, we believe further exploration of per-class class augmentation is a promising practical research direction.
>
> *“The authors should discuss the potential impact of the claim that data is more responsible than the model design in determining invariance, because if true it would imply that researchers may have a stronger responsibility to attend to dataset construction than to modeling when considering potential negative outcomes of a model.”*
>
> We thank the reviewer for pointing out our findings highlight the importance of dataset construction for determining the learned invariances of models. We will include a specific discussion of this finding’s broader implication for the research community.

---

> > ### Comment · Reviewer_dbox · 2021-08-31
> > **no change**
> >
> > Thank you. I am still fully in favor of acceptance and I don't think prescriptions are required---the results stand on their own.

---

### Decision · Program_Chairs · 2021-09-27

**Decision:**

Accept (Poster)

**Comment:**

This paper brought about very divergent opinions from the reviewers — they all agreed that the analyses across the three sections are technically solid, and the topic is of broad interest to the community, but the main points of disagreement were over the novelty of the results, the paper being hard to read since it is packed with results, and whether the paper provides any actionable insights for future modeling approaches. I don’t necessarily view actionable insights as 100% necessary (although they would certainly add value to the paper), but upon reading the paper myself, I agree with reviewer WyYA’s assessment that Sections 3 and 4 are probably the more interesting ones and I wish the authors had dived deeper into them and perhaps had a more cohesive messaging for the paper (it is indeed packed with many, many results which makes it hard to digest). Overall however, I do think some of the results presented in this paper are surprising and will inform ongoing and future research in this field.